# Optimal Environmental Policy in a Dynamic Transboundary Pollution Game: Emission Standards, Taxes, and Permit Trading

**Hao Xu** [1],[*] and **Ming Luo** [2]

1    School of Economics and Management, Southwest Petroleum University, Chengdu 610500, China
2    School of Economics and Management, Southwest Jiaotong University, Chengdu 610031, China; mingluo@my.swjtu.edu.cn
*    Correspondence: xuhao232006@163.com

**Abstract:** Global environmental problems such as transboundary pollution and global warming have been recognized as major issues around the world. In practice, governments of all countries are actively exploring various environmental policies to control pollution. The government needs to consider the impact of neighboring regions when formulating environmental policies, especially in the context of transboundary pollution. However, the above problems are less studied, to bridge this gap and aim at solving problems in existing practices, we consider a differential game model of transboundary pollution control to examine which policy is more effective in promoting environmental quality and social welfare in a dynamic and accumulative global pollution context. Three alternative policy instruments, namely emission standards, emission taxes, and emission permit trading, are considered and compared. The results show that the social welfare of each region is the lowest and the total pollution stock is the highest under the emission tax policy due to the "rent-shifting," "policy-leakage," and "free-riding" effects. Moreover, the realized level of the environmental policy in the Nash equilibrium of the policy game is distorted away from the socially optimal level. The emission standards policy is found to be better than the emission tax policy and characterized by initiating the rent-shifting effect without the policy-leakage effect. Moreover, the pollution stock of two regions is found to be the lowest and the social welfare is found to be the highest under the emission permit trading policy, which is not associated with any of the three effects. Finally, a numerical example is used to illustrate the results, and a sensitivity analysis is performed in the steady state.

**Keywords:** environmental policy; transboundary pollution; emission standards; emission taxes; emission permit trading; differential game

## 1. Introduction

According to the report of the global Environmental Performance Index (EPI) released by Yale University and Columbia University in 2018 ([1]), the world is still far from achieving international environmental goals; the EPI score was reported to be only 46.16. This score reflects the current environmental condition and the implementation effects of environmental policies. The EPI reveals that the environmental quality of the world is improving as compared with the last few decades, but pollution remains severe, especially in developing countries and regions. Environmental pollution originates from the excessive discharge of pollutants by industrial firms. However, due to the pollutant attributes of externality and public goods, firms are reluctant to reduce their emissions or invest in emission abatement activities, resulting in market failure. Therefore, governments have adopted command-and-control or market-based environmental policies, such as emission standards, taxes, permit trading, etc., to govern the environment. As the most important means of pollution control, the above environmental policies have received extensive attention from academia and government [2–4]. Emission standards set quantitative limits on

the permissible amount of specific air pollutants that may be released from specific sources over specific timeframes. As a command-and-control policy, if the pollutant emissions of enterprises exceed this standard, they will face serious environmental penalties [5]. Both emission taxes and emission permit trading are market-based environmental policies. The most significant difference between the two is the uncertainty of emission reduction and emission permit price. Emission tax is a price control policy, the tax rate is set by the policy makers, and the emission reduction is determined by the market. While the emission permit trading is a quantitative policy, the total emission is set by the policy maker, and the trading price is determined by the market mechanism. When the permit price or total emission is set at the place where the marginal abatement cost is equal to the marginal abatement income, the emission tax and emission permit trading can implement the same outcome under the completely competitive market conditions. Scholars have carried out long-term and systematic research on the comparison of the above policies. However, uncertainties in reality make all kinds of policies unable to achieve their theoretical optimal effects, and all kinds of policies need to be reasonably selected according to the specific environment [2,6].

What is more, a typical feature of pollution is that pollutants are released from one region and then migrate to another; this is called transboundary pollution, which often occurs in two or more neighboring regions. Many or all regions generate emissions, and many or all are also suffering from them. A celebrated example is the emission of greenhouse gases (GHG), sulfur dioxide ($SO_2$), and ozone, which cause global warming and environmental degradation [7]. Transboundary pollution not only destroys the ecological environment, but also endangers human health. Outdoor air pollution caused by PM2.5, $SO_2$ and ozone leads to premature death for more than 3 million people worldwide every year [8]. Therefore, in view of the impact of neighboring regions, when the government regulates the environmental policies of local enterprises, it should not only consider the impact of local environmental pollution, but also consider the impact of the behavior of the government and enterprises in adjacent areas, which increases the difficulty of decision-making and may cause new changes in the optimal choice of environmental policies [9]. Most related studies conducted previously were limited to the optimal environmental policy of either local or domestic governments, and some scholars compared the emission taxes, standards, and permit trading policies under the conditions of different market situations and implementation environments [10–12]. However, these studies did not compare the environmental and social welfare effects of the three environmental policies, particularly the conditions of the optimal policy choice for the transboundary pollution problem, which is the main objective of the present study. The consideration of transboundary pollution is unique; for two or more adjacent regions, each region suffers not only from the damage caused by the pollution stock generated by local firms, but also from the pollution stock generated by firms in the neighboring region [13,14]. Therefore, the government of each region will also be affected by the neighboring region when implementing environmental policies.

Two adjacent regions are considered in this study. Each region has a representative firm that competes with the representative firm of the other region and sells homogeneous goods, and the firm in each region will emit pollutants during the production process. Consumers in the two regions can also freely purchase the products produced by the firms in the two regions. The goal of each government is to maximize the local social welfare under the constraint of the total pollution stock of the two regions. Moreover, the two regions are considered to be suffering from the same environmental damage as a result of the total emissions. Thus, differential game models of the optimal choice problem are first formulated in the following three scenarios: emission standards, emission taxes, and emission permit trading. The feedback Nash equilibrium solutions of the total pollution stock and the social welfare of each region are then derived, and the three policies are compared. The results are illustrated with a numerical example.

The main contributions of this paper include the following: (1) Most of the existing studies are limited to the static situation and focus on the relationship between a single

government and enterprises. This paper studies the optimal choice of government environmental policy under the dynamic change of transboundary pollution and pollution capacity; this is the main expansion of and improvement on the existing research; (2) emission standards, emission taxes and emission permit trading were analyzed in our paper, which not only systematically compares the differences in environmental quality of emission standards, emission taxes and emission permits trading environmental policies, but also compares the impact on social welfare in each region, and analyzes the reasons for the differences. It has important practical significance for the control of transboundary pollution.

The main structure of this paper is as follows: Section 1 of the paper introduces the research background and describes the purpose of the work and its significance. Section 2 summarizes the relevant literature and presents the contributions of scholars in this field and the research innovation of this paper. Section 3 provides the game formulation between the government and firms under the condition of transboundary pollution. Subsequently, Section 4 introduces the following scenarios: (i) the emission standards policy in Section 4.1; (ii) the emission tax policy in Section 4.2; (iii) the emission permit trading policy in Section 4.3. Section 4.4 compares two aspects of the three environmental policies, namely the total pollution stock and the social welfare benefits for the government in each region. In Section 5, the results are illustrated with a numerical example. Finally, Section 6 concludes this study with a brief summary and suggests future research directions.

## 2. Literature Review

The predominant strategy used to reduce pollution is environmental policy. Thus, topics related to environmental policy and the comparison of different policies have frequently been examined in the field of environmental economics and management [15–19]. These investigations have tried to establish which type of regulation is desirable in terms of social welfare or environmental quality. Requate [10] first compared the tax and permit policies for an asymmetric quantity duopoly setting. In general, the two policies have no significant difference and neither taxes nor permits result in the social optimum due to imperfect competition; however, for a considerable range of parameters, the permit policy is better than taxes in terms of social welfare. Masoudi and Zaccour [17] compared price-based (taxes) and quantity-based (quotas) environmental policies. One significant distinction of this study is that the authors introduced market uncertainty, and the regulator was considered to have Bayesian learning curve characteristics. The results showed that the emission levels under the tax policy and quota policy were the same, but from the perspective of social welfare, the tax policy was found to be better than the quota policy, which is due to the impact of uncertainty. Feenstra et al. [20] analyzed the effects of emission taxes and standards on environmental and abatement investment in the context of an international duopoly; they proved that among the feedback strategies, emission standards are better than emission taxes, as taxes induce the substitution between capital and pollution input. Hoel and Karp [21] compared the environmental effects of tax and quota policies. When the damages suffered by the government depend on the flow of pollution, the effects of the two policies depend on the discount rate and the environmental decay rate; a higher discount rate and environmental decay rate increase the likelihood of the government's preference of taxes. Lee and Park [22] demonstrated that the differentiation of the product in a duopoly can play a significant role in the policy choice of the government; it was found that when the product difference between firms is large, the consumer surplus under the condition of the emission permit trading policy is lower than that under the condition of the command-and-control policy. Kato [11] considered the environmental policies of taxes and quotas set by the government in a mixed duopoly under both uniform and differentiated situations; it was found that social welfare is the greatest under differentiated emission quotas. Gacia et al. [23] compared the differences between the emission tax and emission permit trading policies under the credible commitment and non-commitment policies of the government. The authors established a Cournot model, and found that the effects of the two policies are the same under the government's commitment to implement

a certain environmental policy, but the tax policy causes less environmental damage under the non-commitment policy. Feichtinger et al. [24] studied the environmental externalities caused by enterprise production and emissions on the basis of oligopoly differential game, and supervised them simultaneously through an emission tax and emission cap policy. The results showed that the green research and development investment curve of enterprises showed an inverted U shape over time and was closely related to the government's environmental policy. Moner-Colonques and Rubio [25] examined the effects that the strategic use of environmental innovation has on environmental policy (taxes and standard) and its welfare implications in a duopoly. The results showed that the strategic behavior of firms is welfare-improving and may induce more environmental innovation than under regulatory commitment only when a tax is used to control pollution and the convexity of investment costs is relatively more important than that of environmental damages.

However, these studies only considered domestic regions, and ignored the impact of pollution emissions from neighboring regions or countries. Therefore, some scholars have investigated the environmental policy issues among two or multiple regions. For example, Ulph [26] considered the choice of environmental policy instruments (taxes or standards) in the context of a model of strategic international trade between countries. The results revealed that if trade is modeled as a one-off Cournot equilibrium, then there is no difference between the two policies for all countries. However, if trade is modeled as a Stackelberg game, then both countries have higher producer surpluses under the emission standard policy. Lai and Hu [27] investigated the import tariff and tax policy instruments between two countries, each of which has one firm producing a differentiated good. The main results indicated that the two countries should subsidize the imported dirty goods in the presence of strong transboundary pollution. On the other hand, if the transboundary pollution is relatively weak, then the countries should set positive tariffs. Glachant et al. [28] proposes a partial equilibrium model with imperfect competition in both the North and South regional and international polluting product markets. They analyzed the impact on technology transfer when the North and South governments set emission quotas non-cooperatively or cooperatively.

A differential game is an effective tool with which to study environmental pollution control problems and analyze the interactions between the strategic behaviors of participants [29]. The dynamic game of transboundary pollution was considered by Dockner and Van Long [30], Jorgensen and Zaccour [31,32], Breton et al. [33], Li [34], Bertinelli et al. [35], and El Ouardighi et al. [36]. Considering the characteristics of transboundary pollution involving two neighboring countries, Dockner and Van Long [30] characterized cooperative and non-cooperative pollution control strategies of the governments of the two countries that maximize the discounted stream of net benefits of a representative consumer, he found that the Markov-perfect equilibrium can be achieved when governments use non-linear strategies. Li [34] established a differential game of the transboundary policy that allows emission permit trading to be carried out between asymmetric regions. The paper mainly found that the optimal cooperative emission rates are lower than the optimal non-cooperative emission rates and the optimal cooperative quantity of purchased/sold emission permits is lower than the optimal non-cooperative quantity of purchased/sold emission permits. Yeung and Petrosyan [37] first presented a cooperative stochastic differential game of transboundary industrial pollution with two features. One is that the pollution can be divided into short- and long-term impacts, another feature is that they explored a payment distribution mechanism. Xu and Tan [38] considered the abatement technology licensing between governments in two adjacent regions; in contrast, the current work considers the optimal environmental policy between the government and firms in two adjacent regions. Marsiglio and Masoudi [39] constructed a transboundary pollution control problem in a two-country differential game, they suggest that a universally homogeneous environmental tax may not be either desirable or optimal in spite of the non-cooperative and the cooperative solutions. Li and Guo [40] developed a dynamic decision model of transboundary basin pollution with emission permits trading and pollution abatement

investment. De Frutos et al. [41] used a differential game model to compare the equilibrium trajectories of the stocks of pollution and cleaner technology as well the regions' welfare. De Frutos and Martín-Herrán [42] analyzed a transboundary pollution differential game and introduced a spatial dimension to capture the geographical relationships among regions. Yanase [43] examined the effects of environmental policy on the total global environment as an international public good with a stock externality, and found that the emission tax game brings about larger strategic distortions than the emission permit game in the absence of global cooperation. Yanase [44] also considered the market structure of enterprises in two countries competing in a third country, and the author analyzed the impacts of emission tax and emission standard policies adopted by the governments of the two countries on the total pollution stock. The results revealed that the emission standards policy is better than the emission tax policy in terms of both the environment and social welfare. Yanase and Kamei [45] develop a two-country differential game model of transboundary pollution control with a continuum of polluting and oligopolistic industries. Governments choose the path of their emission permits and the price of the emission permits is determined by the market-clearing mechanism.

Although the academic community has carried out a number of studies on environmental policy and pollution control, there is still room for further expansion, as follows: (1) in the past, scholars have mainly studied the current situation and problems of environmental policy, but few scholars have studied the optimal environmental policy under transboundary pollution. This research expands and improves upon the existing research; (2) the existing studies did not explicitly consider inter-regional trade in goods or resources that cause pollution; and (3) discussions on the evaluation of alternative environmental policies, such as emission taxes, emission standards, or marketable emission permits, are absent from the literature. These characteristics reflect the main variations between the present study and previous research.

## 3. Parameter Description, Assumptions and Methods

### 3.1. Parameter Description and Assumptions

For convenience, the main parameters of this paper are described in Table 1.

**Table 1.** Notations and descriptions.

| Notation | Description |
|----------|-------------|
| $\alpha_m$ | The positive constant parameter measuring the reservation price, $m = i, j$ |
| $a$ | The market size of each region |
| $\chi_m$ | The quantity of the product purchased by consumers in each region, $m = i, j$ |
| $\gamma_m$ | The cost coefficient of emission reduction, $m = i, j$ |
| $\delta$ | The natural decay rate |
| $\rho$ | The discount rate |
| $d_m$ | The damage parameter, $m = i, j$ |
| $x(0)$ | The initial level of the pollution stock, $x(0) = x_0 > 0$ |
| $p(t)$ | The product price at time $t$ |
| $q_m(t)$ | The output of the firm at time $t$, $m = i, j$ |
| $r_m(t)$ | The pollution abatement level at time $t$, $m = i, j$ |
| $x(t)$ | The pollution stock in the two regions at time $t$ |
| $\pi_m(t)$ | The instantaneous profit of the firm at time $t$, $m = i, j$ |
| $\theta_m(t)$ | The emission standard set by the government in each region at time $t$, $m = i, j$ |
| $\tau_m(t)$ | The emission tax rate at time $t$, $m = i, j$ |
| $\overline{E}_m(t)$ | The emission quota of the firm assigned by the government in each region at time $t$, $m = i, j$ |
| $V_m$ | The value function of each region, $m = i, j$ |

A transboundary pollution model comprising two adjacent regions or countries, namely region $i$ and region $j$, is considered. In each region, consumers purchase the same homogeneous product in the common market formed by the two regions. According to Glachant et al. [28], the demand function of consumers in each region is assumed to be $\chi_m = \alpha_m - \frac{p}{2}$, $m = i, j$, where $\chi_m$ represents the quantity of goods purchased by consumers

in region $i$ or region $j$, $\alpha_m$ is a positive constant parameter measuring the reservation price (alternatively, it is a measure of the market size), and $p$ is the price of the product, which is the same in both regions. Accordingly, the consumer surplus in each region can be calculated as follows.

$$CS_m = \int_p^{2\alpha_m} \left(\alpha_m - \frac{p}{2}\right) dp = \frac{(2\alpha_m - p)^2}{4} \tag{1}$$

According to the consumer demand function, the inverse demand function of products in the common market can be obtained as follows.

$$p(c_i + c_j) = \alpha_i + \alpha_j - \chi_i - \chi_j \tag{2}$$

The product is produced by two firms, one in each region, which compete "à la Cournot." In the usual manner, it is assumed that the market size in each region is the same, so $\alpha_i = \alpha_j = \alpha$. Production generates pollution, and, for simplicity, each unit of output for the firm at time $t$ is considered to create one unit of pollutant, $q_i(t)$ or $q_j(t)$ [6]. The firm can reduce emissions by end-of-pipe management, and each firm located in region $i$ or region $j$ is supposed to have a quadratic total abatement cost of $\frac{\gamma_m}{2} r_m{}^2(t)$, $m = i, j$, where $r_m(t)$ is the total pollution abatement level for each firm at time $t$, and $\gamma_m$ is the cost coefficient of emission reduction. The total cost of firm $m$ is additively separable and given by $cq_m + \frac{\gamma_m}{2} r_m{}^2(t)$, where $c$ is the unit cost of production. The proximity of the two regions leads to the cross-flow of pollutants emitted by the firms, and the focus of this paper is the stock of pollutants, which evolves according to the following differential equation [46]:

$$\dot{x}(t) = q_i(t) + q_j(t) - r_i(t) - r_j(t) - \delta x(t), \ x(0) = x_0, \ x(t) \geq 0 \tag{3}$$

where $x(t)$ represents the pollution stock and $\delta > 0$ is the natural decay rate of pollution. An advantage of the application of a differential game to problems of pollution is the opportunity to model damage caused by the stock of accumulated pollution [46,47]. According to Masoudi and Zaccour [17], Breton [33], and Menezes and Pereira [46], the damage caused by the stock of pollution for region $m$ at time $t$ can be measured by $d_m x(t)$, where $d_m > 0$ is the damage parameter. The damage $d_m x(t)$ means that each region suffers not only from the damage caused by the emissions of its own firm, but also from the damage caused by the emissions of the firm in the neighboring region. The objective of the firm in each region is to choose its output and reductions to maximize its profits. On the other hand, to reduce the environmental damage caused by the emissions of the firm, the government in each region chooses the optimal policy with the aim of maximizing social welfare, which is defined as the sum of the consumer surplus and producer surplus minus the environmental damages subject to differential Equation (3). Three alternative policy regimes are considered, namely emission standards, emission taxes, and emission permit trading. It is assumed that the government has more information about the total environmental quality and its impact on local welfare than do the firms. Specifically, the pollution stock and environmental damage of each region in Equation (3) can be observed by the governments, but it cannot be judged by the firms. Therefore, the firm cannot predict policy information, such as the emission standards, emission tax rates, or initial emission permits set by the government. These assumptions indicate that the local government can obtain information about the firm's feedback strategy and the damage caused by the total pollution in the two regions to formulate the optimal temporal path under each environmental policy to maximize social welfare.

*3.2. Methods*

3.2.1. Differential Game

Differential games have distinct advantages in order to represent the interdependencies among time, strategic behavior and participants in mathematical models in the fields of environmental economics and optimal pollution control. Firstly, an advantage of differential games in applications to problems of pollution is the opportunity to model damage caused by the stock of accumulated pollution (Jørgensen et al., 2010). Therefore,

one of the key assumptions of this paper is that emissions by either region contribute to the stock of pollution and the two regions face the same pollution stock. Secondly, the strategy structure of the differential game solution reflects the interaction of the behaviors between the participants. By establishing the Hamilton–Jacobi–Bellman equation (HJB), the Markov-perfect Nash equilibrium (MPNE) is obtained. Under this equilibrium, the participants not only consider the dynamic changes of state variables, but also adjust their own strategies based on the decision-making choices of other participants, and finally achieve time consistency and subgame perfection. However, for multiple state equations or non-linear situations, the differential game cannot achieve its analytical solution, meaning that numerical analysis can be used to verify the results, which is its limitation

### 3.2.2. Stackelberg Game

The Stackelberg leadership model is a strategic game in economics in which the leader firm moves first and then the follower firms move sequentially. In our paper, it is assumed that the government has more information about the total environmental quality and its impact on local welfare than do the firms. Specifically, the pollution stock and environmental damage of each region in Equation (3) can be observed by the governments (Leaders), but it cannot be judged by the firms (Followers). Therefore, the firm cannot predict policy information, such as the emission standards, emission tax rates, or initial emission permits set by the government. These assumptions indicate that the local government can obtain information about the firm's feedback strategy and the damage caused by the total pollution in the two regions to formulate the optimal temporal path under each environmental policy to maximize social welfare. In particular, a three-stage Stackelberg game is considered in which: (1) the government in each region sets the standards/taxes/quotas to maximize social welfare, (2) firms determine the abatement level, and (3) market competition occurs (firms determine output). We can use backward induction to solve this problem and guarantee the equilibrium solution concept is in perfect subgame equilibrium [48,49].

## 4. Dynamic Games of Environmental Policies

### 4.1. Emission Standards (Scenario S)

Under the policy of emission standards, we consider the case in which the government in each region pre-commits to the level of environmental policy. In particular, a three-stage game is considered in which: (1) the government in each region sets the standards to maximize social welfare, (2) firms determine the abatement level, and (3) market competition occurs (firms determine output). The equilibrium solution concept is in perfect subgame equilibrium with backward induction. The output of the firm in region $m$ is the function of the standard and abatement level: $q_m(t) = \theta_m(t) + r_m(t)$, where $\theta_m(t)$ is the standard set by the local government in each region at time $t$. According to the inverse demand function and market clearing condition $\chi_i + \chi_j = q_i + q_j$, $p = 2a - q_i - q_j$ can be obtained. Then, the profit of each firm in region $i$ and region $j$ at time $t$ can be expressed as follows.

$$\underset{q_i, r_i}{\pi_i(t)} = \left[\alpha + \alpha - q_i(t) - q_j(t)\right]q_i(t) - cq_i(t) - \frac{\gamma_i}{2}r_i^2(t) \tag{4}$$

$$\underset{q_j, r_j}{\pi_j(t)} = \left[\alpha + \alpha - q_i(t) - q_j(t)\right]q_j(t) - cq_j(t) - \frac{\gamma_j}{2}r_j^2(t) \tag{5}$$

In the last stage, namely the market competition stage, the firm in each region chooses its output to maximize Equations (4) and (5); $\pi_m(t)$ is concave in $q_m(t)$ because $\partial^2 \pi_m(t)/\partial q_m^2(t) = -2 < 0$, $m = i, j$. The optimal outputs $q_i(t)$ and $q_j(t)$ can then be obtained as follows (the subscript $S$ represents the emission standard situation, and the time $t$ is omitted for simplicity):

$$q_{iS}\left(\theta_i, \theta_j\right) = \frac{2(a + \gamma_i\theta_i) + \gamma_j\left(2a + \gamma_i\theta_i - \theta_j\right)}{3 + 2\gamma_i + 2\gamma_j + \gamma_i\gamma_j}, \quad q_{jS}\left(\theta_i, \theta_j\right) = \frac{2\left(a + \gamma_j\theta_j\right) + \gamma_i\left(a + \gamma_j\theta_j\right)}{3 + 2\gamma_i + 2\gamma_j + \gamma_i\gamma_j} \tag{6}$$

where, for notational simplicity, $a = \alpha - c$ is set, and $a$ measures the market size. In the second stage, the firm in each region determines its abatement level to maximize its payoffs; $\pi_m(t)$ is concave in $q_m(t)$ because $\partial^2 \pi_m(t)/\partial r_m{}^2(t) = -\gamma_i < 0$, $m = i, j$. Solving these problems yields the equilibrium abatement levels, as follows.

$$r_{iS}(\theta_i, \theta_j) = \frac{2a - 3\theta_i + \gamma_j(2a - 2\theta_i - \theta_j)}{3 + 2\gamma_i + 2\gamma_j + \gamma_i\gamma_j}, \quad r_{jS}(\theta_i, \theta_j) = \frac{2a - 3\theta_j + \gamma_i(2a - 2\theta_j - \theta_i)}{3 + 2\gamma_i + 2\gamma_j + \gamma_i\gamma_j} \quad (7)$$

The condition to ensure that the results are non-negative is $a > \max\{(2\theta_i + \theta_j)/2, (2\theta_j + \theta_i)/2\}$. The output and abatement level of each firm depend on the allowable emissions in two regions; $\partial q_{iS}(\theta_i, \theta_j)/\partial \theta_i > 0$, $\partial q_{jS}(\theta_i, \theta_j)/\partial \theta_i < 0$, which means that when the emission standard level in region $i$ is more restrictive (i.e., a decrease in $\theta_i$), the output of firm in region $i$ will decrease correspondingly, while the output of the firm in region $j$ will increase. This phenomenon is called the "rent-shifting" effect [41,50], i.e., local strict environmental policies will reduce the output of the local firm, but will simultaneously increase the output of the firm in the neighboring region. From the reaction function of the pollution abatement level of the firm in region $i$, $\partial r_{iS}(\theta_i, \theta_j)/\partial \theta_i < 0$, $\partial r_{jS}(\theta_i, \theta_j)/\partial \theta_i < 0$, and the net emissions $\partial E_{iS}(\theta_i, \theta_j)/\partial \theta_i > 0$ and $\partial E_{jS}(\theta_i, \theta_j)/\partial \theta_i = 0$ can be obtained. This means that a stricter emission standard in region $i$ will increase the level of pollution abatement of the firm in region $i$, and will also increase the pollution abatement level of the firm in the neighboring region; however, a stricter emission standard in region $i$ will have no effect on the net emissions in the neighboring region $j$.

In the first stage, the government in each region selects the standard that maximizes the discounted present value of social welfare. Thus, the objective function and the constraint condition under the emission standards policy can be expressed as follows:

$$\max_{\theta_m} = \int_0^\infty e^{-\rho t}\{CS_m(t) + \pi_m(t) - d_m x(t)\}dt,$$

$$s.t. \ \dot{x}(t) = q_i(t) + q_j(t) - r_i(t) - r_j(t) - \delta x(t), \ x(0) = x_0, \ x(t) \geq 0,$$

where the parameter $\rho(0 < \rho < 1)$ is the discount rate. The problem can be solved by using the Hamilton–Jacobi–Bellman (HJB) equation [29]. By substituting Equations (6) and (7) into the government's objective function, and via symmetry ($d_i = d_j$, $\gamma_i = \gamma_j$), the HJB equations of region $i$ and region $j$ are obtained, respectively, as follows:

$$\rho V_{iS}(x) = \max_{\theta_i} \left\{ \frac{(4a + \gamma\theta_i + \gamma\theta_j)^2}{4(3+\gamma)^2} - \frac{(2a(1+\gamma) + \gamma\theta_i(2+\gamma) - \gamma\theta_j)(-2a(1+\gamma) + \gamma\theta_i + \gamma\theta_j)}{(3+\gamma)(3+4\gamma+\gamma^2)} \right.$$
$$\left. - \frac{\gamma(-2a(1+\gamma) + (3+2\gamma)\theta_i + \gamma\theta_j)^2}{2(3+4\gamma+\gamma^2)^2} - dx(t) + V'_{iS}(x)(\theta_i + \theta_j - \delta x(t)) \right\} \quad (8)$$

$$\rho V_{jS}(x) = \max_{\theta_j} \left\{ \frac{(4a + \gamma\theta_i + \gamma\theta_j)^2}{4(3+\gamma)^2} - \frac{(2a(1+\gamma) + \gamma\theta_j(2+\gamma) - \gamma\theta_i)(-2a(1+\gamma) + \gamma\theta_i + \gamma\theta_j)}{(3+\gamma)(3+4\gamma+\gamma^2)} \right.$$
$$\left. - \frac{\gamma(-2a(1+\gamma) + (3+2\gamma)\theta_j + \gamma\theta_i)^2}{2(3+4\gamma+\gamma^2)^2} - dx(t) + V'_{jS}(x)(\theta_i + \theta_j - \delta x(t)) \right\} \quad (9)$$

where $V_{iS}(x)$ and $V_{jS}(x)$ are the value functions of region $i$ and region $j$, respectively. Following the solutions provided by Dockner et al. [29], the feedback Nash equilibrium solutions of the optimal standards $\theta_i^*$ and $\theta_j^*$, the total pollution stock trajectory of both regions $x_S^*(t)$, and the optimal social welfare of both regions $V_{iS}^*(x)$ and $V_{jS}^*(x)$ can be obtained, as given by Proposition 1.

**Proposition 1.** *The feedback Nash equilibrium solutions of the optimal standards, the total pollution stock trajectory of both regions, and the optimal social welfare of each region are given, respectively, by the following.*

$$\theta_i^* = \theta_j^* = \frac{2a\gamma(5 + 5\gamma + \gamma^2)(\rho + \delta) - d(1+\gamma)(3+\gamma)^2}{(9 + 10\gamma + 2\gamma^2)(\rho + \delta)} \quad (10)$$

$$x_S^*(t) = (x_0 - x_{SS}^*)e^{-\delta t} + x_{SS}^* \tag{11}$$

$$V_{iS}^*(x) = V_{jS}^*(x) = \frac{d^2(3+\gamma)^2(27+56\gamma+35\gamma^2+6\gamma^3)}{2\gamma\rho(9+10\gamma+2\gamma^2)^2(\rho+\delta)^2} - \frac{4ad\gamma(81+175\gamma+131\gamma^2+39\gamma^3+4\gamma^4)}{\gamma\rho(9+10\gamma+2\gamma^2)^2(\rho+\delta)^2}$$
$$+ \frac{4a^2\gamma(36+80\gamma+62\gamma^2+19\gamma^3+2\gamma^4)}{\gamma\rho(9+10\gamma+2\gamma^2)^2(\rho+\delta)^2} - \frac{d}{\rho+\delta}x_S^* \tag{12}$$

The proof for Proposition 1 is presented in the Appendix A.

In Equations (10)–(12), $x_{SS}^* = \frac{2(2a\gamma(5+5\gamma+\gamma^2)(\rho+\delta)-d(1+\gamma)(3+\gamma)^2)}{\gamma\delta(9+10\gamma+2\gamma^2)(\rho+\delta)}$ represents the steady state of the total pollution stock in the two regions, and $x_0$ is the initial condition of the pollution stock. When $x_0 < x_{SS}^*$, $\partial x_S^*/\partial t > 0$, it means that if the initial pollution stock of the regions is less than that of the steady state, the evolution of the pollution stock is an accumulative process; when $x_0 > x_{SS}^*$, $\partial x_S^*/\partial t < 0$, the evolution of the pollution stock is a dissipative process; when $x_0 = x_{SS}^*$, $\partial x_S^*/\partial t = 0$, the pollution stock is constant.

Under the government's optimal emission standards policy, $q_{iS}^*$, $q_{jS}^*$, $r_{iS}^*$, $r_{jS}^*$ are, respectively, the equilibrium outputs and emission reductions of the firms in the two regions under the optimal emission standards, which are defined, respectively, as follows.

$$q_{iS}^* = q_{jS}^* = \frac{(3+4\gamma+\gamma^2)(2a(\rho+\delta)-d)}{(9+10\gamma+2\gamma^2)(\rho+\delta)} \tag{13}$$

$$r_{iS}^* = r_{jS}^* = \frac{3d(3+4\gamma+\gamma^2) - 2a\gamma(2+\gamma)(\rho+\delta)}{\gamma(9+10\gamma+2\gamma^2)(\rho+\delta)} \tag{14}$$

*4.2. Emission Taxes (Scenario T)*

Similar to the description of Section 4.1, in the third stage of emission taxes, the firms in the two regions choose their outputs to maximize their respective payoffs. Given the emission tax rate $\tau_m(t)$, the profit function of the firm in each region is as follows.

$$\max_{q_m, r_m} \pi_m(t) = \left(\alpha + \alpha - q_i(t) - q_j(t)\right)q_m(t) - cq_m(t) - \frac{\gamma_m}{2}r_m(t)^2 - \tau_m(t)(q_m(t) - r_m(t)) \tag{15}$$

$\pi_m(t)$ is concave in $q_m(t)$ because $\partial^2\pi_m(t)/\partial q_m^2(t) = -2 < 0$, $m = i, j$. Given the emission tax rate, the first-order conditions give the following equilibrium output level of each firm and the total outputs:

$$q_{iT}(\tau_i, \tau_j) = \frac{2a - 2\tau_i + \tau_j}{3}, \quad q_{jT}(\tau_i, \tau_j) = \frac{2a - 2\tau_j + \tau_i}{3} \tag{16}$$

where $a = \alpha - c$. In the second stage, firms choose their abatement level to maximize their respective payoffs in Equation (15). $\pi_m(t)$ is concave in $r_m(t)$ because $\partial^2\pi_m(t)/\partial r_m^2(t) = -\gamma_m < 0$, $m = i, j$. The first-order conditions give the following equilibrium pollution abatement levels as a function of the tax.

$$r_{iT}(\tau_i, \tau_j) = \frac{\tau_i}{\gamma_i}, \quad r_{jT}(\tau_i, \tau_j) = \frac{\tau_j}{\gamma_j} \tag{17}$$

The condition to ensure that the results are non-negative is $a > \max\{(2\tau_i - \tau_j)/2, (2\tau_j - \tau_i)/2\}$. Based on Equation (15), $\partial q_{iT}(\tau_i, \tau_j)/\partial\tau_i < 0$, $\partial q_{jT}(\tau_i, \tau_j)/\partial\tau_i > 0$, $\partial q_{iT}(\tau_i, \tau_j)/\partial\tau_j > 0$, and $\partial q_{jT}(\tau_i, \tau_j)/\partial\tau_j < 0$. This means that the increase in the emission tax rate in the local region (i.e., region *i*) reduces the output of the firm in the local region, whereas it increases the output of the firm in the neighboring region (i.e., region *j*), i.e., the rent-shifting phenomenon. Regarding the abatement levels of the firms, $\partial r_i(\tau_i, \tau_j)/\partial\tau_i > 0$, $\partial r_j(\tau_i, \tau_j)/\partial\tau_j > 0$, the emission abatement levels of the firm in region *i* increase with the increase in $\tau_i$, but the firm in region *j* is not affected by the increase in $\tau_i$. Regarding the net emissions of each region, $\partial E_{iT}(\tau_i, \tau_j)/\partial\tau_i < 0$, $\partial E_{jT}(\tau_i, \tau_j)/\partial\tau_i > 0$, $\partial E_{iT}(\tau_i, \tau_j)/\partial\tau_j > 0$, and $\partial E_{jT}(\tau_i, \tau_j)/\partial\tau_j < 0$; this indicates that with the increase in the emission tax rate imposed by the government of the local region (region *i*), the net emissions of the local region

(i.e., region $i$) decrease, while those of the neighboring region (i.e., region $j$) increase. This can be interpreted as the "policy leakage" of the environmental policy, which is different from that under the emission standards policy. Under the emission standards policy, the strict environmental policy in the local region (region $i$) has no impact on the net emissions of the firm in the neighboring region. Based on the response function of each firm, the government maximizes social welfare subject to the dynamics of $x$. Analogously to the emission standards policy game, the value functions of region $i$ and region $j$ are denoted, respectively, as $V_{iT}(x)$ and $V_{jT}(x)$. The HJB equations then become the following.

$$\rho V_{iT}(x) = \max_{\tau_i} \left\{ \begin{array}{l} \frac{(-4a+\tau_i+\tau_j)^2}{36} + \frac{-9\tau_i^2+2\gamma\left(-2\tau_i^2-\tau_i(2a+\tau_j)+(2a+\tau_j)^2\right)}{18\gamma_i} - dx(t) \\[10pt] + V'_{iT}(x)\left(\frac{1}{3}(4a-\tau_i-\tau_j) - \frac{\tau_i+\tau_j}{\gamma} - \delta x(t)\right) \end{array} \right\} \tag{18}$$

$$\rho V_{jT}(x) = \max_{\tau_j} \left\{ \begin{array}{l} \frac{(-4a+\tau_i+\tau_j)^2}{36} + \frac{-9\tau_j^2+2\gamma_j\left(4a^2+\tau_i^2+\tau_i(4a-\tau_j)-2a\tau_j-2\tau_j^2\right)}{18\gamma_j} \\[10pt] -d_jx(t) + V'_{jT}(x)\left(\frac{1}{3}(4a-\tau_i-\tau_j) - \frac{\tau_i+\tau_j}{\gamma_i} - \delta x(t)\right) \end{array} \right\} \tag{19}$$

The feedback Nash equilibrium solutions of the optimal emission taxes of each region $\tau_i^*$ and $\tau_j^*$, the total pollution stock trajectory of both regions $x_T^*(t)$, and the optimal social welfare of each region $V_{iT}^*(x)$ and $V_{jT}^*(x)$ can therefore be obtained, as given by Proposition 2.

**Proposition 2.** *The feedback Nash equilibrium solutions of the optimal emission taxes, the total pollution stock trajectory of both regions, and the optimal social welfare of each region are given, respectively, by*

$$\tau_i^* = \tau_j^* = \frac{3d(3+\gamma) - 4a\gamma(\rho+\delta)}{(9+4\gamma)(\rho+\delta)} \tag{20}$$

$$x_T^*(t) = (x_0 - x_{TS}^*)e^{-\delta t} + x_{TS}^* \tag{21}$$

$$\begin{array}{l} V_{iT}^* = V_{jT}^* = -\frac{d}{\rho+\delta}x_T^* + \frac{d^2(3+\gamma)^2(27+14\gamma)-4ad\gamma\left(81+73\gamma+16\gamma^2\right)(\rho+\delta)}{2\gamma(9+4\gamma)^2\rho(\rho+\delta)^2} \\[12pt] \qquad\quad + \frac{16a^2\gamma\left(9+8\gamma+2\gamma^2\right)(\rho+\delta)^2}{2\gamma\rho(9+4\gamma)^2(\rho+\delta)^2} \end{array} \tag{22}$$

*where* $x_{TS}^* = \frac{2\left(2a\gamma(5+2\gamma)(\rho+\delta)-d(3+\gamma)^2\right)}{\gamma\delta(9+4\gamma)(\rho+\delta)}$ *represents the steady state of the total pollution stock in the two regions.*

The proof is presented in the Appendix B.

Moreover, $q_{iT}^*$, $q_{jT}^*$, $r_{iT}^*$, $r_{jT}^*$ are the equilibrium outputs and emission reductions, respectively, of the firms in two regions under the optimal emission taxes, and are defined, respectively, as follows.

$$q_{iT}^* = q_{jT}^* = \frac{2a(3+2\gamma)(\rho+\delta) - d(3+\gamma)}{(9+4\gamma)(\rho+\delta)} \tag{23}$$

$$r_{iT}^* = r_{jT}^* = \frac{3d(3+\gamma) - 4a\gamma(\rho+\delta)}{(9+4\gamma)(\rho+\delta)} \tag{24}$$

*4.3. Tradable Emission Permits (Scenario P)*

In this scenario, $\overline{E}_i(t)$ and $\overline{E}_j(t)$ are the emission quotas of the firms assigned by the government in each region at time $t$. Given the emission quotas, the governments also allow the firms to trade emission permits in the emission trading market. The emission trading market is assumed to be a centralized competitive market in which emission trading occurs via the market clearing price. Thus, if the net demand of the firm in region $m$ is defined as $D_m = q_m - r_m - \overline{E}_m$, the total net demand of emission permits is zero at the

market equilibrium $D_i + D_j = 0$. This also implies that both firms do not have market power in the emission market, and they therefore behave as price takers in the centralized competitive emission trading market in which firms trade permits at the realization of the market clearing price [19].

In the third stage, the firm in each region chooses its outputs to maximize the following function.

$$\max_{q_m, r_m} \pi_m(t) = \left( \alpha + \alpha - q_i(t) - q_j(t) \right) q_m(t) - cq_m - \frac{1}{2} \gamma_m r_m(t)^2 - \lambda_m \left( q_m(t) - r_m(t) - \overline{E}_m(t) \right) \quad (25)$$

$\pi_m(t)$ is concave in $q_m(t)$ because $\partial^2 \pi_m(t) / \partial q_m^2(t) = -2 < 0$, $m = i, j$. The equilibrium output as the function of the emission permit price $\lambda_m$ is obtained as follows:

$$q_{iP}(\lambda_i, \lambda_j) = \frac{1}{3}(2a - 2\lambda_i + \lambda_j), \ q_{jP}(\lambda_i, \lambda_j) = \frac{1}{3}(2a + \lambda_i - 2\lambda_j), \quad (26)$$

where $a = \alpha - c$. Subscript $T$ is employed to denote the equilibrium under the tradable permits policy. In the second stage, firms choose their abatement levels to maximize their payoffs. $\pi_m(t)$ is concave in $r_m(t)$ because $\partial^2 \pi_m(t) / \partial r_m^2(t) = -\gamma_m < 0$. The optimal abatement levels can be determined as follows.

$$r_{iP}(\lambda_i) = \frac{\lambda_i}{\gamma_i}, \ r_{jP}(\lambda_j) = \frac{\lambda_j}{\gamma_j} \quad (27)$$

To ensure that the results are non-negative, $a > \max\left\{ (2\lambda_i - \lambda_j)/2, \ (2\lambda_j - \lambda_i)/2 \right\}$. The emission trading market is assumed to be a centralized competitive market in which emission trading occurs via the market clearing price, and where $\overline{E}_i + \overline{E}_j = \frac{1}{3}(2a - 2\lambda_i + \lambda_j) - \frac{\lambda_i}{\gamma_i} + \frac{1}{3}(2a - 2\lambda_j + \lambda_i) - \frac{\lambda_j}{\gamma_j}$, $\lambda_i = \lambda_j = \lambda$. The following can then be obtained.

$$\lambda = \frac{\gamma_i \gamma_j \left( 4a - 3\overline{E}_i - 3\overline{E}_j \right)}{3\gamma_i + 3\gamma_j + 2\gamma_i \gamma_j} \quad (28)$$

By substituting Equation (28) into Equations (26) and (27), the outputs and emission abatement levels are obtained as functions of the emission quotas, as follows.

$$q_{iP}\left( \overline{E}_i, \overline{E}_j \right) = \frac{2a\gamma_j + \gamma_i \left( 2a + \overline{E}_i \gamma_j + \overline{E}_j \gamma_j \right)}{3\gamma_i + 3\gamma_j + 2\gamma_i \gamma_j}, \ q_{jP}\left( \overline{E}_i, \overline{E}_j \right) = \frac{2a\gamma_j + \gamma_i \left( 2a + \overline{E}_i \gamma_j + \overline{E}_j \gamma_j \right)}{3\gamma_i + 3\gamma_j + 2\gamma_i \gamma_j} \quad (29)$$

$$r_{iP}\left( \overline{E}_i, \overline{E}_j \right) = \frac{\left( 4a - 3\overline{E}_i - 3\overline{E}_j \right) \gamma_j}{3\gamma_i + 3\gamma_j + 2\gamma_i \gamma_j}, \ r_{jP}\left( \overline{E}_i, \overline{E}_j \right) = \frac{\left( 4a - 3\overline{E}_i - 3\overline{E}_j \right) \gamma_i}{3\gamma_i + 3\gamma_j + 2\gamma_i \gamma_j} \quad (30)$$

$\partial q_{iP}\left( \overline{E}_i, \overline{E}_j \right) / \partial \overline{E}_i > 0, \partial q_{jP}\left( \overline{E}_i, \overline{E}_j \right) / \partial \overline{E}_i > 0, \partial q_{iP}\left( \overline{E}_i, \overline{E}_j \right) / \partial \overline{E}_j > 0$ and $\partial q_{jP}\left( \overline{E}_i, \overline{E}_j \right) / \partial \overline{E}_j > 0$; these relations state that when the local government reduces the initial emission quota assigned to the local firm (strict policy), the output of the firm will decrease, and the output of the firm in the neighboring region will also decrease accordingly. This means that there is no rent-shifting effect under the emission permit trading policy. Regarding the abatement levels of the firms, $\partial r_{iP}\left( \overline{E}_i, \overline{E}_j \right) / \partial \overline{E}_i < 0, \partial r_{jP}\left( \overline{E}_i, \overline{E}_j \right) / \partial \overline{E}_i < 0$ and $\partial r_{iP}\left( \overline{E}_i, \overline{E}_j \right) / \partial \overline{E}_j < 0, \partial r_{jP}\left( \overline{E}_i, \overline{E}_j \right) / \partial \overline{E}_j < 0$. It can also be demonstrated that $\partial E_{iP}\left( \overline{E}_i, \overline{E}_j \right) / \partial \overline{E}_i > 0, \partial E_{iP}\left( \overline{E}_i, \overline{E}_j \right) / \partial \overline{E}_j > 0, \partial E_{jP}\left( \overline{E}_i, \overline{E}_j \right) / \partial \overline{E}_j > 0$, and $\partial E_{jP}\left( \overline{E}_i, \overline{E}_j \right) / \partial \overline{E}_i > 0$. These relations state that the reduction in the initial emission permit quota in the local region will reduce the net emissions of firms in both regions, and there is no policy leakage. Based on the response function of each firm, the government maximizes social welfare subject to the dynamics of $x$. The HJB equations of the government in each region can then be obtained as follows.

$$\rho V_{iP}(x) = \max_{e_i} \left\{ \frac{\left( 4a + \gamma \overline{E}_i + \gamma \overline{E}_j \right)^2}{4(3+\gamma)^2} + \frac{16a^2(2+\gamma) - \gamma \overline{E}_i^2(27+10\gamma) - 8a\gamma \overline{E}_j}{8(3+\gamma)^2} \right.$$
$$\left. + \frac{\gamma(9+2\gamma)\overline{E}_j^2 + 2\gamma \overline{E}_{ii}\left( 4a(5+2\gamma) - (9+4\gamma)\overline{E}_j \right)}{8(3+\gamma)^2} - dx(t) + V_{iP}'(x)\left( \overline{E}_i + \overline{E}_j - \delta x(t) \right) \right\} \quad (31)$$

$$\rho V_{jP}(x) = \max_{e_j} \left\{ \frac{\left(4a+\gamma\overline{E}_i+\gamma\overline{E}_j\right)^2}{4(3+\gamma)^2} + \frac{16a^2(2+\gamma)+\gamma\overline{E}_i^2(9+2\gamma)+8a\gamma\overline{E}_j(5+2\gamma)}{8(3+\gamma)^2} \right.$$
$$\left. - \frac{\gamma(27+10\gamma)\overline{E}_j^2-2\gamma\overline{E}_i\left(4a-\overline{E}_j(9+4\gamma)\right)}{8(3+\gamma)^2} - dx(t) + V'_{jP}(x)\left(\overline{E}_i+\overline{E}_j-\delta x(t)\right) \right\}$$

(32)

Similar to the previous analysis, the feedback Nash equilibrium solutions of the optimal initial permit quotas $\overline{E}_i^*$ and $\overline{E}_j^*$, the total pollution stock trajectory of both regions $x_P^*(t)$, and the optimal social welfare of each region $V_{iP}^*(x)$ and $V_{jP}^*(x)$ can be obtained, as given by Proposition 3.

**Proposition 3.** *The feedback Nash equilibrium solutions of the optimal emission permit quotas, the total pollution stock trajectory of both regions, and the optimal social welfare of each region are given, respectively, by*

$$\overline{E}_i^* = \overline{E}_j^* = \frac{2\left(a\gamma(7+2\gamma)(\rho+\delta)-d(3+\gamma)^2\right)}{\gamma(18+5\gamma)(\rho+\delta)}$$

(33)

$$x_P^*(t) = (x_0 - x_{PS}^*)e^{-\rho t} + x_{PS}^*$$

(34)

$$V_{iT}^* = V_{jT}^* = \frac{2d^2(3+\gamma)^2(27+8\gamma)-4ad\gamma\left(135+77\gamma+11\gamma^2\right)(\rho+\delta)}{\gamma\rho(18+5\gamma)^2(\rho+\delta)^2} +$$
$$\frac{2a^2\gamma\left(144+83\gamma+12\gamma^2\right)}{\gamma\rho(18+5\gamma)^2} - \frac{d}{\rho+\delta}x_P^*$$

(35)

*where* $x_{PS}^* = \frac{4\left(a\gamma(7+2\gamma)(\rho+\delta)-d(3+\gamma)^2\right)}{\gamma\delta(18+5\gamma)(\rho+\delta)}$ *expresses the steady state of the total pollution stock in the two regions.*

The proof is provided in Appendix C.

Moreover, $q_{iP}^*$, $q_{jP}^*$, $r_{iP}^*$, $r_{jP}^*$ are the equilibrium outputs and abatement levels of the firms, respectively, in the two regions under the emission permit trading policy.

$$q_{iP}^* = q_{jP}^* = \frac{2(3+\gamma)(2a(\rho+\delta)-d)}{(18+5\gamma)(\rho+\delta)}$$

(36)

$$r_{iP}^* = r_{jP}^* = \frac{2(3d(3+\gamma)-a\gamma(\rho+\delta))}{\gamma(18+5\gamma)(\rho+\delta)}$$

(37)

*4.4. Comparing Emission Standards, Taxes, and Permit Trading*

The feedback Nash equilibrium results of the total pollution stock and the social welfare of each region under the three environmental policies were obtained. In this section, the different effects of the three policies comprising pollution stock are compared in terms of the environmental quality and social welfare of each region.

(1) Difference in the trajectory and steady state of the pollution stock

From Propositions 1–3, the different results of the optimal trajectories of the pollution stock under the three environmental policies can be derived as follows.

$$x_T^* - x_S^* = \frac{2\left(1-e^{-\delta t}\right)(3+\gamma)\left(\left(9+9\gamma+2\gamma^2\right)+2a\gamma(\rho+\delta)\right)}{(9+4\gamma)(9+10\gamma+2\gamma^2)(\rho+\delta)} > 0$$

(38)

$$x_T^* - x_P^* = \frac{2(3+\gamma)\left(1-e^{-\delta t}\right)(3d(3+\gamma)+2a(9+2\gamma)(\rho+\delta))}{(9+4\gamma)(18+5\gamma)(\rho+\delta)} > 0$$

(39)

$$x_S^* - x_P^* = \frac{2(3+\gamma)^3\left(1-e^{-\delta t}\right)(2a(\rho+\delta)-d)}{(18+5\gamma)(9+10\gamma+2\gamma^2)(\rho+\delta)} > 0$$

(40)

$$x_{TS}^* - x_{SS}^* = \frac{2(3+\gamma)\left(\left(9+9\gamma+2\gamma^2\right)+2a\gamma(\rho+\delta)\right)}{\delta(9+4\gamma)(9+10\gamma+2\gamma^2)(\rho+\delta)} > 0$$

(41)

$$x_{TS}^* - x_{PS}^* = \frac{2(3+\gamma)[3d(3+\gamma) + 2a(9+2\gamma)(\rho+\delta)]}{\delta(9+4\gamma)(18+5\gamma)(\rho+\delta)} > 0 \tag{42}$$

$$x_{SS}^* - x_{PS}^* = \frac{2(3+\gamma)^3[2a(\rho+\delta) - d]}{\delta(18+5\gamma)(9+10\gamma+2\gamma^2)(\rho+\delta)} > 0 \tag{43}$$

From Equation (35), $2a(\rho+\delta) - d > 0$ was determined. Moreover, regarding the environmental decay rate, $0 < \delta < 1$ and $1 - e^{-\delta t} > 0$, based on which the following proposition is obtained.

**Proposition 4.** *The equilibrium trajectory and steady state of the pollution stock under the emission tax policy are larger than those under the emission standards and emission permit trading policies. Furthermore, the equilibrium trajectory and steady state of the pollution stock are the lowest under the emission permit trading policy*, i.e., $x_T^* > x_S^* > x_P^*$ and $x_{TS}^* > x_{SS}^* > x_{PS}^*$.

Proposition 4 demonstrates that when the government in each region implements environmental policy under the condition of transboundary pollution, the total pollution stock of the two regions under the emission tax policy is the highest from the perspective of the environmental effect. Furthermore, among the three policies, the total pollution stock under the emission permit trading policy is the lowest. Our conclusion is partially consistent with the views of Ulph [26] and Yanase [44]. For example, Ulph considered the choice of environmental policy instruments (taxes or standards) in the context of a model of strategic international trade between countries. The results showed that if trade is modeled as a Stackelberg game, then both countries have higher producer surpluses under the emission standard policy. Yanase [44] considered the market structure of enterprises in two countries competing in a third country, and he analyzed the impacts of tax and standard policies adopted by the governments of the two countries on the total pollution stock. The results revealed that the emission standards policy is better than the emission tax policy in environmental quality. However, these scholars did not participate in the analysis of emission permit trading policy, and did not compare the comprehensive impact on the environment and social welfare. In addition, they also did not analyze the deep-seated reasons for the above results. Our conclusion will fill these research gaps.

The reasons for this can be explained as follows. As presented in Sections 4.1–4.3, under the emission tax policy, when the government of the local region (i.e., region *i*) increases the emission tax rate (the policy tends to be strict), the firm in the local region will decrease its outputs, whereas the firm in the neighboring region (i.e., region *j*) will increase its outputs accordingly; this is the rent-shifting effect. Moreover, the increase in the emission tax rate by the local government will decrease the net emissions of the local region, but will lead to an increase in the net emissions in the neighboring region, resulting in policy leakage. This means that when the government attempts to improve the environmental quality by increasing the emission tax rate, "free-riding" behavior will occur in the neighboring region. For these reasons, when determining the emission tax rate, the government in each region will predict and consider the free-riding behavior of the other government. Therefore, the emission tax rates of both sides will be lower than the optimal level, which will lead to increases in the net emissions and pollution stock. Some of the literatures on environmental economics have revealed that local governments tend to lower the standards of environmental regulation to attract scarce working capital and enterprises to enter the local area, resulting in inferior competition between governments and intensifying environmental pollution [51,52].

Similarly, when the emission standard level in the local region (region *i*) is more restrictive (i.e., a decrease in emission standards), the firm in the local region will decrease its outputs, but the firm in the neighboring region will increase its outputs. However, as compared with the emission tax policy, the rent-shifting effect is relatively weak. The reason for this is that the reduction in emission standards in the local region increases the emission abatement levels of not only the local firm, but also those of the firm in the neighboring

region. Therefore, a portion of the increasing emissions in the neighboring region is offset, and no policy leakage will occur. Taking into account these reactions, the rent-shifting effect will be stronger under the emission tax policy by relaxing the environmental policy. In other words, compared with the emission standards policy, the environmental policy of the local government under the emission tax policy is more relaxed, thereby leading to a larger pollution stock.

Finally, in terms of the emission permit trading policy, the reduction in the initial emission quota of the firm in the local region (i.e., region *i*) will reduce not only the output of the local firm, but also that of the firm in the neighboring region (i.e., region *j*). Consequently, the rent-shifting effect does not exist. Furthermore, the reduction in the emission quota in the local region will lead to the increase in the emission abatement levels in both regions. Therefore, the reduction in the emission quota in the local region will increase the net emissions of the firms in both the local and neighboring regions, and no policy leakage problem will exist. There will also be no free-riding, which would cause the governments of the two regions to relax their environmental policies. According to the preceding analysis, among the three policies, the pollution stock under the emission permit trading policy will be the lowest, and the environmental performance will be the best.

(2)    Difference in social welfare under three policies

From Propositions 1–3, the following different results in the optimal welfare under the three policies ($V_{iS}^*(x) = V_{jS}^*(x) = V_S^*(x)$, $V_{iT}^*(x) = V_{jT}^*(x) = V_T^*(x)$, and $V_{iP}^*(x) = V_{jP}^*(x) = V_P^*(x)$) can be obtained.

$$V_S^*(x) - V_T^*(x) = \frac{(54+51\gamma+10\gamma^2)[d(9+9\gamma+2\gamma^2)+2a\gamma(\rho+\delta)]^2}{2\rho(9+4\gamma)^2(9+10\gamma+2\gamma^2)^2(\rho+\delta)^2} - \frac{d}{\rho+\delta}(x_S^* - x_T^*) > 0 \tag{44}$$

From Equation (37), $x_S^* - x_T^* < 0$ can be determined; then, $V_S^*(x) - V_T^*(x) > 0$.

$$V_P^*(x) - V_S^*(x) = \frac{(2a(\rho+\delta)-d)(3+\gamma)^3}{2\rho(\rho+\delta)^2(18+5\gamma)^2(2\gamma^2+10\gamma+9)^2}[d(324 + 441\gamma + 177\gamma^2 + 22\gamma^3) - 2a\gamma(5\gamma + 2\gamma^2)(\rho + \delta) + 18a\gamma(\rho + \delta)] - \frac{d}{\rho+\delta}(x_P^* - x_S^*) \tag{45}$$

From Equation (36), $3d(3 + \gamma) - 4a\gamma(\rho + \delta) > 0$. For $\gamma > 0$, via the multiplication of both sides of the inequality by $\frac{5\gamma+2\gamma^2}{2}$, the following inequality can be obtained.

$$\frac{3}{2}d(3 + \gamma)(5\gamma + 2\gamma^2) > 2a\gamma(\rho + \delta)(5\gamma + 2\gamma^2) \tag{46}$$

From Equation (45), the following results can be obtained.

$$d(324 + 441\gamma + 177\gamma^2 + 22\gamma^3) - 2a\gamma(5\gamma + 2\gamma^2)(\rho + \delta) > 0 \tag{47}$$

Finally, from Equations (39) and (35), $x_P^* - x_S^* < 0$ and $2a(\rho + \delta) - d > 0$. Moreover, the result of Equation (44) is $V_P^*(x) - V_S^*(x) > 0$. Then, from $V_S^*(x) - V_T^*(x) > 0$ and $V_P^*(x) - V_S^*(x) > 0$, Proposition 5 on the difference in social welfare can be obtained as follows.

**Proposition 5.** *When the government in each region implements an environmental policy under transboundary pollution, the social welfare of each region under the emission tax policy will be less than that under the emission standards policy, $V_S^*(x) - V_T^*(x) > 0$. Moreover, the social welfare of each region under the emission standards policy will be less than that under the emission permit trading policy, $V_P^*(x) - V_S^*(x) > 0$. In other words, the social welfare under the emission permit trading policy is the highest, that under the emission standards policy is the second highest, and that under the emission tax policy is the lowest.*

Proposition 5 reveals that, from the perspective of the social welfare of each region, the emission tax policy is the worst, while the emission permit trading policy is the best.

According to the previous hypothesis, the social welfare for region *i* or region *j* is the sum of the consumer surplus ($CS_m$) and the profit of the firm in each region minus environmental damage. It is known that the consumer surplus is the same under the three environmental policies because the market price of commodities is the same. Therefore, the social welfare of each region primarily depends on the differences in the profits of the firm and the environmental damage suffered by each region. Due to the rent-shifting and policy-leakage effects, the environmental policy under the emission tax policy is looser than that under the emission standards policy; furthermore, due to the policy-leakage effect, the environmental policy under the emission standards policy is looser than that under the emission permit trading policy. This means that, compared with the other two policies, the firm under the emission tax policy has the highest output and the lowest emission abatement level (the abatement cost is the lowest). However, the response of the firm in each region to the policy of the neighboring government must be considered.

For the local region (region *i*), when the government adopts the loose policy, the output $q_{iT}$ increases and the abatement level $r_{iT}$ decreases with the decrease in the emission tax rate $\tau_i$. Similarly, the output $q_{iS}$ or $q_{iP}$ increases and the abatement level $r_{iS}$ or $r_{iP}$ decreases with the increase in the emission standard $\theta_i$ or emission quota $\overline{E}_i$. However, when considering the policy of the neighboring region, the output $q_{iT}$ decreases with the decrease in the emission tax rate $\tau_j$, but there is no impact on the abatement level $r_{iT}$. Moreover, output $q_{iS}$ decreases and $r_{iS}$ decreases with the increase in the emission standard $\theta_j$ of region *j*, and the output $q_{iP}$ increases and the abatement level $r_{iP}$ decreases with the increase in the initial quota $\overline{E}_j$ of region *j*. This demonstrates that a decline in the emission tax rate in the neighboring region lowers the output of the firm in the local region, and for the emission standards policy, an increase in the emission standard level in the neighboring region lowers not only the output of the local firm, but also its abatement level, which reduces the abatement cost of the local firm. Finally, an increase in the emission quota in the neighboring region increases the output of the local firm and simultaneously lowers its abatement level. The preceding analysis demonstrates that the difference between the output and abatement level (abatement cost) under the emission permit trading policy is the greatest, that under the emission standards policy is the second greatest, and that under the emission tax policy is the least.

According to Proposition 4, the total pollution stock in the two regions is the highest under the emission tax policy, and is the lowest under the emission trading policy (the environmental damage suffered by each region is the lowest). Therefore, Proposition 5 can be interpreted from the preceding analysis.

## 5. Analysis of the Equilibrium Results

### 5.1. Equilibrium Trajectories

In this section, the equilibrium results of each region under the three policies and under different initial pollution stocks are analyzed. Reference is made to several previous studies [6,53], and representative parameter settings are considered as a basic example to illustrate the findings. The basic parameter settings are given as follows: the market size $a = 60$, the abatement of the firm in each region $\gamma_i = \gamma_j = 1$, the pollution damage suffered by each region $d_i = d_j = 4$, the discount rate $\rho = 0.05$, and the environmental decay rate $\delta = 0.1$.

Figure 1 presents the equilibrium trajectories of the pollution stock when the total initial pollution stock $x_0$ in the environment is 800 and 100, respectively.

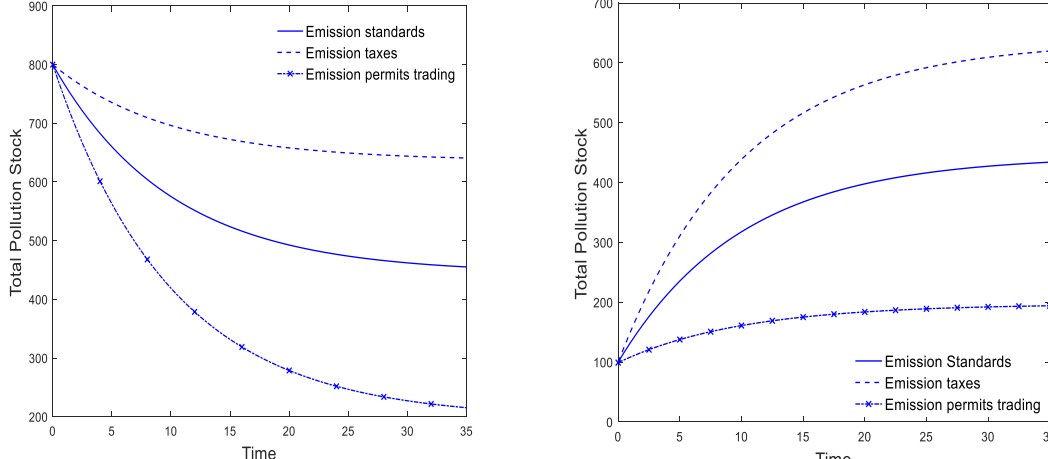

**Figure 1.** The equilibrium trajectories of the pollution stock in the three scenarios under the initial. pollution stock $x_0 = 800$ (**left**), and $x_0 = 100$ (**right**).

Figure 1 presents the numerical results of the three scenarios over time. The total pollution stock of the two regions under the emission tax policy was found to be the highest among the three policies. Via the comparison of the emission standard and emission permit trading policies, it was found that the total pollution stock under the emission standards policy is higher than that under the emission permit trading policy. Moreover, when the initial pollution stock of the two regions is higher than the pollution stock under the steady state, the trajectories first descend and then converge to stationary levels over time. On the contrary, if the initial pollution stock of the two regions is less than the pollution stock under the steady state, the trajectories first ascend and then converge to stationary levels over time. Figure 2 exhibits the revenue trajectories of each region under the three policies when the initial pollution stock $x_0$ is 800 and 100. The emission permit trading policy was found to always be more beneficial than the emission standards policy, and both were found to be more beneficial than the emission tax policy. Thus, in terms of both social welfare and environmental quality, the emission permit trading policy is always better than the emission standard and emission tax policies.

### 5.2. Steady-State Equilibrium Results

Section 5.1 shows that the Markov-perfect Nash equilibrium of the regions converge to the stationary levels over time. In this section, we examine the impact of changes in environmental damage coefficients, market size and natural decay rate on total pollution stock, and social welfare in each region in a steady-state situation. Figure 3a shows that in the steady state, the total pollution stock of the two regions decreases as the environment damage parameter of region $i$ increases. Figure 3b,c shows the impact of changes in the damage parameter of region $i$ on the social welfare of each region. An increase in the damage parameter of region $i$ will increase the damage by pollution stock, thereby reducing the output of the firm in the local region. For region $j$, however, the social welfare increases as the damage parameter of region $i$ increases because the total pollution stock is reduced due to the reduction in the output of the firm in region $i$.

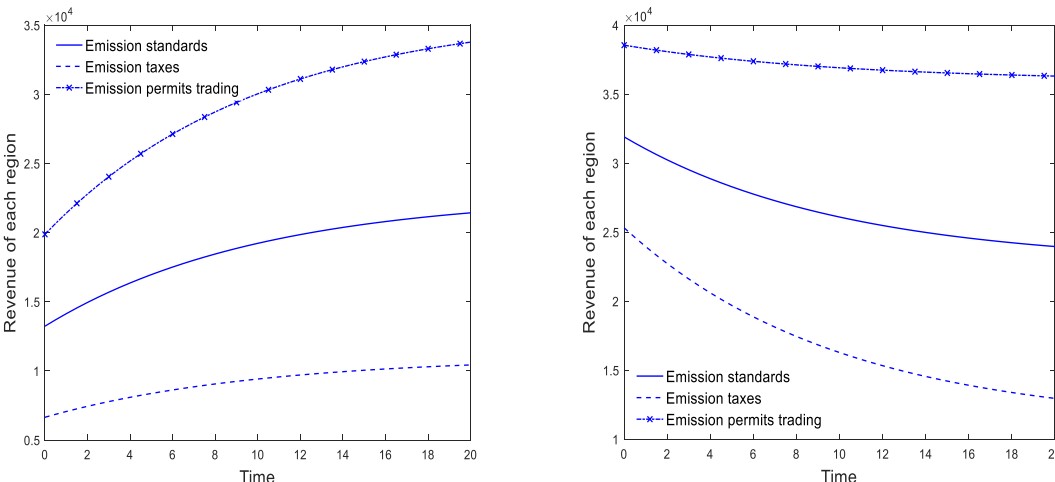

**Figure 2.** The equilibrium trajectories of the revenues in the three scenarios under the initial pollution. stock $x_0 = 800$ (**left**), and $x_0 = 100$ (**right**).

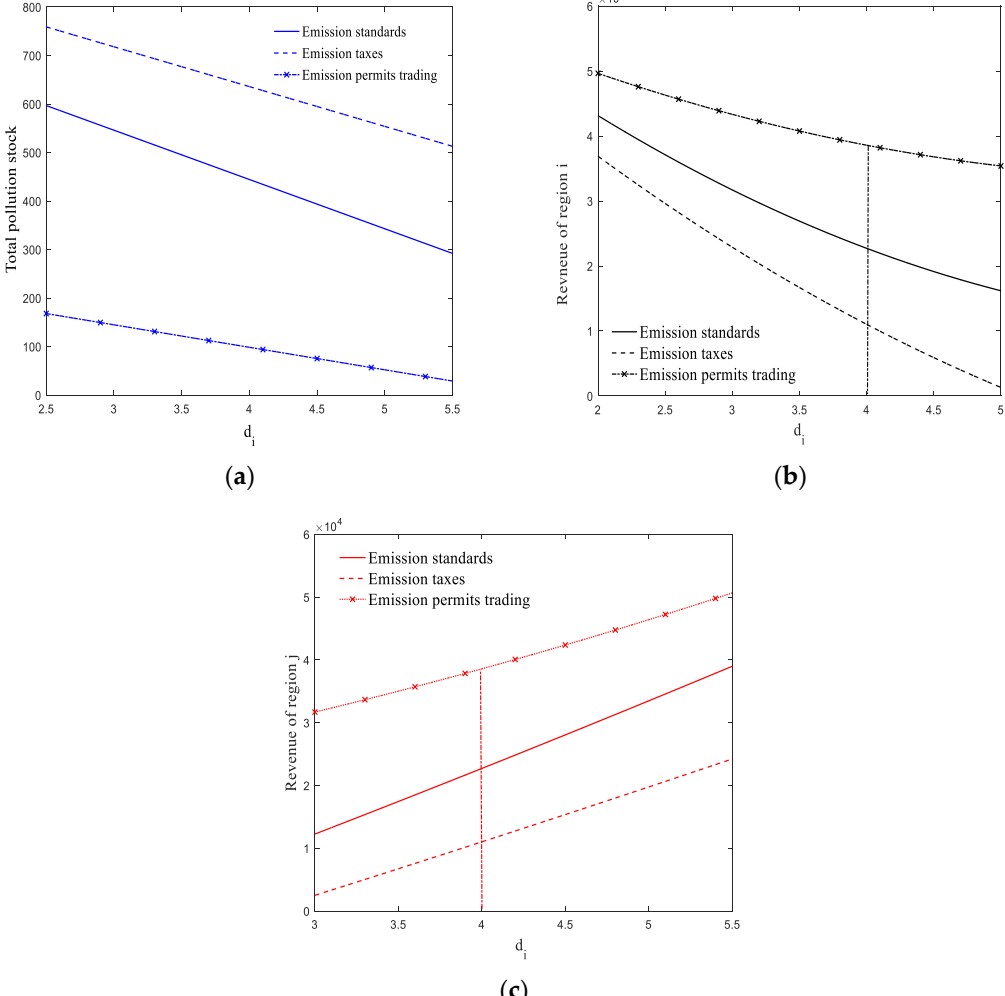

**Figure 3.** Steady-state equilibrium and damage parameter $d_i$; (**a**) Steady-state equilibrium pollution stock; (**b**) Steady-state net revenue of region $i$; (**c**) Steady-state net revenue of region $j$.

Figure 4 shows the impact of changes in market size $a_i$ on the total pollution stock and social welfare. It can be seen from Figure 4a that with the expansion of the market size of region $i$, the total pollution stock of two regions increases due to the increases in

the output of the firm and the total pollution stock. Although the total pollution stock of the two regions has increased, the increase in producer surplus in region *i* causes social welfare to increase with the expansion of the market size $a_i$. Compared with region *i*, due to the increase in pollution stock, region *j* suffers more damage without increasing producer surplus, meaning that its social welfare will be reduced.

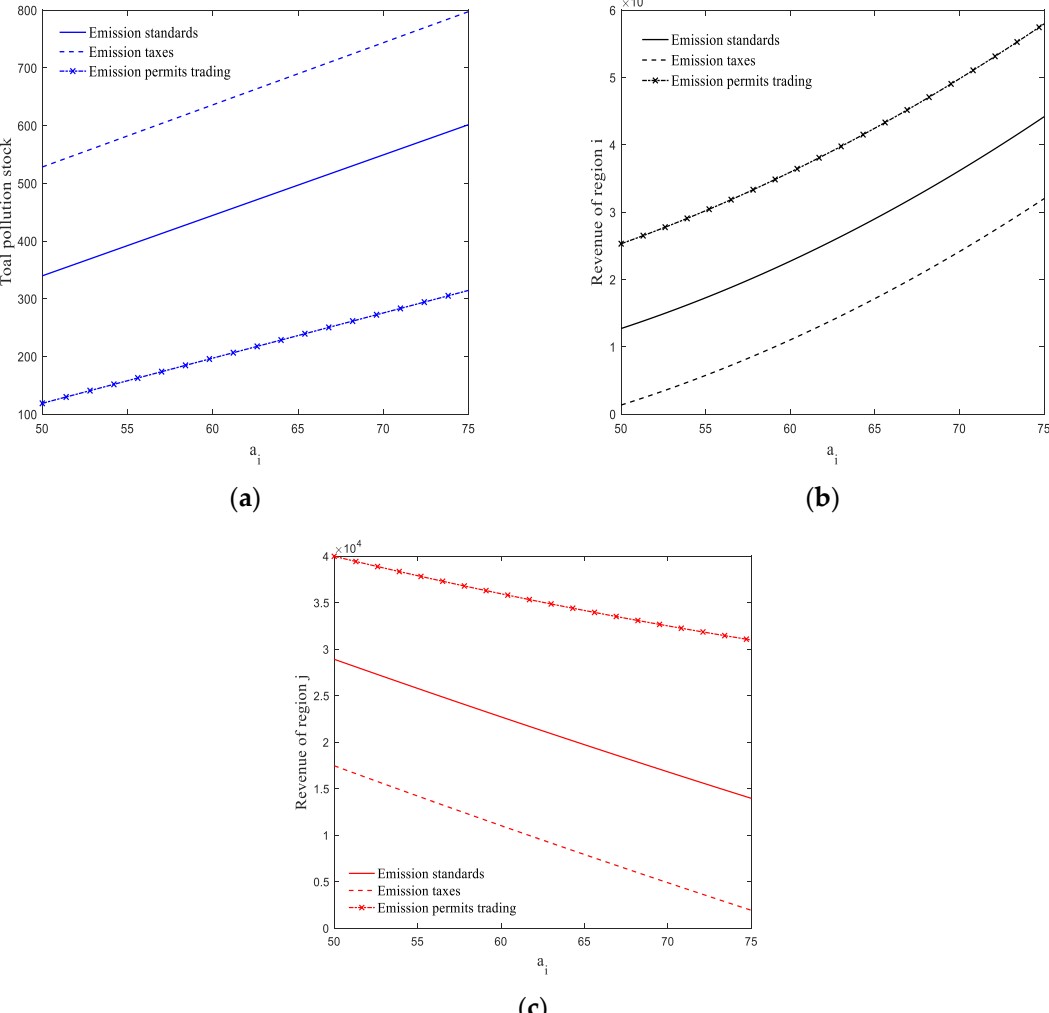

**Figure 4.** Steady-state equilibrium and the market size $a_i$; (**a**) Steady-state equilibrium pollution stock; (**b**) Steady-state net revenue of region *i*; (**c**) Steady-state net revenue of region *j*.

Finally, the influence of the natural decay rate on the steady-state equilibrium results for the two regions is assessed. Figure 5a shows that the total pollution stock of the two regions decreases with the increase in $\delta$, and as the natural decay rate increases, the difference between the three environmental policies becomes smaller and smaller. Figure 5b also shows that when the natural decay rate gradually increases, the social welfare of the two regions under the three environmental policies becomes higher and higher and the welfare difference becomes smaller and smaller.

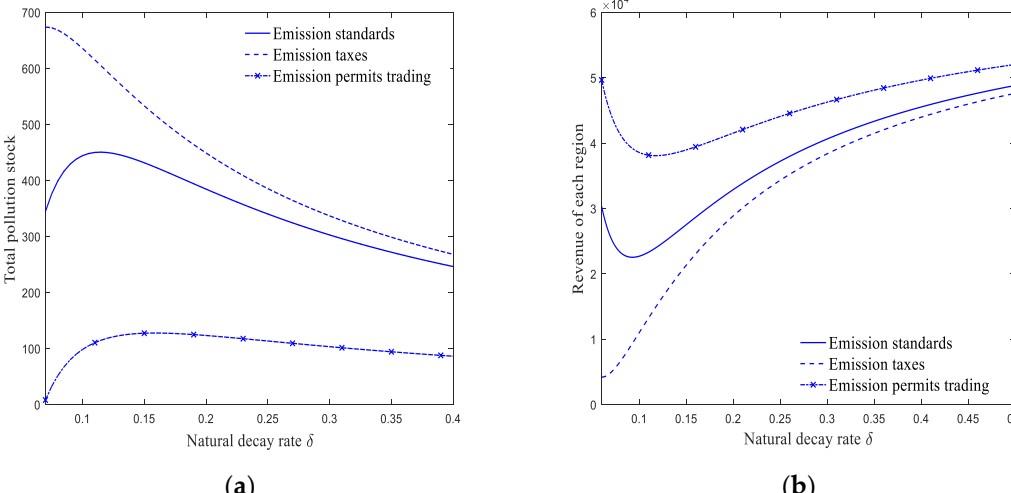

**Figure 5.** Steady-state equilibrium and the natural decay rate $\delta$; (**a**) Steady-state equilibrium pollution stock; (**b**) Steady-state net revenue of each region.

## 6. Conclusions and Applications

### 6.1. Conclusions

The literature on environmental policy usually concentrates on a single region or country. However, a substantial amount of environmental pollution in the world is caused not only by firms in a domestic region, but also by firms in neighboring regions. Therefore, governments will also be affected by neighboring regions when creating environmental policies. In this paper, a transboundary pollution problem was investigated in which there is a Stackelberg game between firms and their local governments, and the government in each region chooses one of three different environmental policies, namely emission standards, emission taxes, and emission permit trading. The feedback Nash equilibrium solutions in the three scenarios were derived, and the three environmental policies were compared from the perspectives of environmental quality and social welfare. The results revealed that: (1) when the emission abatement cost coefficient of the firms in the two regions is the same, the two regions suffer the same environmental damage. Due to the rent-shifting, policy-leakage, and free-riding effects, the total pollution stock was found to be the highest and the social welfare of each region was found to be the lowest under the emission tax policy. The emission standards policy was found to be better than the emission tax policy due to the presence of the rent-shifting effect but the absence of the policy-leakage effect. Finally, the pollution stock of the two regions was found to be the lowest and the social welfare was found to be the highest under the emission permit trading policy, which does not have any of the three effects; (2) the dynamic trajectories of the equilibrium results and the sensitivity analysis of the parameters in the steady state are discussed numerically, and the basic results reveal that when the initial pollution stock of the total regions is higher (lower) than the pollution stock in the steady state, the trajectory first decreases (rises) and converges to the steady level with the passage of time. In addition, the trajectory of the social welfare curve is similar to that of the pollution stock. With the initial pollution stock higher or lower than that the steady state, the social welfare of each region will increase or decrease to the steady-state level accordingly. (3) We analyzed the impact of changes in parameters such as the environmental damage coefficient, market size, and natural decay rate on the total pollution stock and social welfare in each region under steady-state conditions.

In order to fully recognize the significance of the environmental policy under transboundary pollution and improve the total environmental quality, the government should mainly focus on the following four aspects: first, for some countries or regions with serious cross-border pollution, the implementation of regional emissions trading policies may be a

better environmental governance model. For example, in order to alleviate greenhouse gas emissions, the EU implemented the European Union Emissions Trading System (EUETS), which has a significant inhibitory effect on the emissions of carbon dioxide and other air pollutants in the European Union [50]. Second, although the research conclusion of this paper states that the emission tax has the worst effect on environment and welfare, for some developing countries such as China or India, the level of marketization is relatively low, so it can still be combined with the emission tax policy and the emission standard policy to control the pollution.

### 6.2. Limitations and Prospects

The present work discussed the differential game problem of transboundary pollution. However, the difference in the market structure was not considered, which may also yield interesting and comparable results. For example, it would be interesting to know how the environmental policy would change if the market structure was different. In addition, the results are based on the traditional set-up, such as the costate equation for pollution stock and the pollutants damage. Particularly in recent years, a single source of emissions is typically comprised of multiple pollutants which also lead to regional and global negative externalities in reality. Finally, this paper assumes that the government utilizes a grandfathering approach for the initial allocation of costless permits to the firms. More and more governments, however, are now using auctions to deal with the initial allocation of emission rights [54,55]. These issues represent possible extensions of the present study.

**Author Contributions:** Conceptualization, H.X. and M.L.; methodology, H.X.; software, H.X.; validation, H.X. and M.L.; formal analysis, M.L.; resources, H.X.; writing—original draft preparation, H.X.; writing—review and editing, M.L.; visualization, H.X.; supervision, M.L.; project administration, H.X.; funding acquisition, H.X. All authors have read and agreed to the published version of the manuscript.

**Funding:** This research was funded by [National Natural Science Foundation of China] (Grant No. 71571149).

**Institutional Review Board Statement:** Not applicable.

**Informed Consent Statement:** Not applicable.

**Data Availability Statement:** Not applicable.

**Acknowledgments:** The authors express their gratitude to the anonymous reviewers and editors for their helpful comments about this paper.

**Conflicts of Interest:** The authors declare no conflict of interest.

### Appendix A. For Proposition 1

The first-order condition of $\theta_i^*$, $\theta_j^*$ from Equations (7) and (8) are as follows:

$$\theta_i^* = \frac{V_{iS}'(1+\gamma)(3+\gamma)^2 + 2a\gamma(5+5\gamma+\gamma^2)}{\gamma(9+10\gamma+2\gamma^2)} \tag{A1}$$

$$\theta_j^* = \frac{V_{jS}'(1+\gamma)(3+\gamma)^2 + 2a\gamma(5+5\gamma+\gamma^2)}{\gamma(9+10\gamma+2\gamma^2)} \tag{A2}$$

From symmetry we have $V_{iS}'(x) = V_{jS}'(x) = V_S'(x)$, substituting (A1) and (A2) into (7) and (8), we then attain the simplified HJB equation:

$$\rho \cdot V_S(x) = \frac{\left(2a+V_S'(x)\right)^2(1+\gamma)^2(3+\gamma)^2}{(9+10\gamma+2\gamma^2)^2} - \frac{\left(V_S'(x)\right)^2(9+4\gamma)\left(3+4\gamma+\gamma^2\right)^2}{2\gamma(9+10\gamma+2\gamma^2)^2}$$

$$+ \frac{4a\gamma V_S'(x)(3+\gamma)^2\left(3+5\gamma+2\gamma^2\right)-4a^2\gamma\left(18+32\gamma+18\gamma^2+3\gamma^3\right)}{2\gamma(9+10\gamma+2\gamma^2)^2} - dx(t) \qquad (A3)$$

$$+ V_S'(x)\left(\frac{V_S'(x)(1+\gamma)(3+\gamma)^2+2a\gamma\left(5+5\gamma+\gamma^2\right)}{\gamma(9+10\gamma+2\gamma^2)} - \delta x(t)\right)$$

where $V_S(x) = V_{iS}(x) = V_{jS}(x)$, from Equation (3), we guess the form of the value function $V_S(x)$ to be linear in $x$, that is,

$$V_{iS}(x) = V_{jS}(x) = V_S(x) = A_S x + B_S \qquad (A4)$$

where $A_S$ and $B_S$ are constant coefficients. Then, we obtain these constant coefficients to verify that our guess is correct, that means the exact expressions of the value functions are linear. Differentiating Equation (A4) with respect to $x$ and substituting into (A3), we can derive $A_S$ and $B_S$ as follows:

$$A_S = -\frac{d}{\rho + \delta} \qquad (A5)$$

$$B_S = \frac{d^2(3+\gamma)^2\left(27+56\gamma+35\gamma^2+6\gamma^3\right)}{2\gamma(9+10\gamma+2\gamma^2)^2\rho(\rho+\delta)^2} - \frac{4ad\gamma\left(81+175\gamma+131\gamma^2+39\gamma^3+4\gamma^4\right)}{2\gamma(9+10\gamma+2\gamma^2)^2\rho(\rho+\delta)}$$

$$+ \frac{4a^2\gamma\left(36+80\gamma+62\gamma^2+19\gamma^3+2\gamma^4\right)}{2\gamma(9+10\gamma+2\gamma^2)^2\rho} \qquad (A6)$$

Substituting (A5) (A6) into $V_S'(x)$ and from (A1) and (A2), we can determine the feedback Nash equilibrium standards $\theta_i^*$, $\theta_j^*$. By substituting $\theta_i^*$ and $\theta_j^*$ into Equations (5) and (6), we have the optimal output and emission reduction in the firm in each region as follows:

$$q_{iS}^* = q_{jS}^* = \frac{(3+4\gamma+\gamma^2)(2a(\rho+\delta)-d)}{(9+10\gamma+2\gamma^2)(\rho+\delta)} \qquad (A7)$$

$$r_{iS}^* = r_{jS}^* = \frac{3d(3+4\gamma+\gamma^2)-2a\gamma(2+\gamma)(\rho+\delta)}{\gamma(9+10\gamma+2\gamma^2)(\rho+\delta)} \qquad (A8)$$

Substituting Equations (A7) and (A8) into (3) and the steady state of the pollution stock faced by two regions, denoted by $x_{SS}^*$, is determined by setting $\dot{x}_S(t) = 0$:

$$x_{SS}^* = \frac{2\left(2a\gamma\left(5+5\gamma+\gamma^2\right)(\rho+\delta)-d(1+\gamma)(3+\gamma)^2\right)}{\gamma\delta(9+10\gamma+2\gamma^2)(\rho+\delta)}$$

Then, solving the ordinary differential equation, we calculate the following optimal trajectory of the pollution stock:

$$x_S^*(t) = (x_0 - x_{SS}^*)e^{-\rho t} + x_{SS}^*$$

## Appendix B. For Proposition 2

Differentiating the right-hand sides of Equations (17) and (18) with respect to $\tau_i$, $\tau_j$, we can attain the equilibrium emission tax rate as follows:

$$\tau_i = -\frac{-(18+7\gamma)V_{iT}'(x)+\gamma\left(-8a+V_{jT}'(x)\right)}{18+8\gamma} \qquad (A9)$$

$$\tau_j = -\frac{-(18+7\gamma)V_{jT}'(x)+\gamma\left(-8a+V_{iT}'(x)\right)}{18+8\gamma} \qquad (A10)$$

By substituting (A9) and (A10) into (17) and (18), and by symmetry $V_{iT}(x) = V_{jT}(x) = V_T(x)$, we have

$$\rho V_T(x) = -\frac{12aV_T'(x)(3+\gamma) + \left(V_T'(x)\right)^2(3+\gamma)^2(9+2\gamma) - 16a^2\lambda\left(9+8\gamma+2\gamma^2\right)}{2\gamma(9+4\gamma)^2}$$
$$-dx(t) + V_T'(x)\left(\frac{2V'(x)(3+\gamma)^2 + 4a\gamma(5+2\gamma)}{\gamma(9+4\gamma)} - \delta x(t)\right) \tag{A11}$$

From (A11), we conjecture that the structure of (A11) can be regarded as linear value function:

$$V_{iT}(x) = V_{jT}(x) = V_T(x) = A_T x + B_T \tag{A12}$$

Substituting $V_T(x)$ and $V_T'(x)$ into (A11), we can determine the coefficients $A_T$ and $B_T$ as follows:

$$A_T = -\frac{d}{\rho+\delta} \tag{A13}$$

$$B_T = \frac{d^2(3+\gamma)^2(27+14\gamma) - 4ad\gamma(81+73\gamma+16\gamma^2)(\rho+\delta) + 16a^2\gamma(9+8\gamma+2\gamma^2)(\rho+\delta)^2}{2\gamma(9+4\gamma)^2\rho(\rho+\delta)^2} \tag{A14}$$

Substituting (A13) and (A14) into $V_T'(x)$ and referring to (A9) and (A10), we obtain the equilibrium emission taxes $\tau_i^*$, $\tau_j^*$. By substituting $\tau_i^*$ and $\tau_j^*$ into Equations (15) and (16), we attain the optimal output and emission abatement level of the firm in each region as follows:

$$q_{iT}^* = q_{jT}^* = \frac{2a(3+2\gamma)(\rho+\delta) - d(3+\gamma)}{(9+4\gamma)(\rho+\delta)} \tag{A15}$$

$$r_{iT}^* = r_{jT}^* = \frac{3d(3+\gamma) - 4a\gamma(\rho+\delta)}{(9+4\gamma)(\rho+\delta)} \tag{A16}$$

We can attain the steady state of the pollution stock by substituting Equations (A15) and (A16) into (3), and by setting $\dot{x}_T(t) = 0$:

$$x_{TS}^* = \frac{2\left(2a\gamma(5+2\gamma)(\rho+\delta) - d(3+\gamma)^2\right)}{\gamma\delta(9+4\gamma)(\rho+\delta)}$$

Then, solving the ordinary differential equation, we calculate the following optimal trajectory of the pollution stock:

$$x_T^*(t) = (x_0 - x_{TS}^*)e^{-\rho t} + x_{TS}^*$$

## Appendix C. For Proposition 3

From the first-order condition of Equations (30) and (31) we obtain the optimal emission quota:

$$\overline{E}_i = V_{iP}'(x) + \frac{\gamma(4a + \gamma\overline{E}_i + \gamma\overline{E}_j)}{2(3+\gamma)^2} + \frac{-2\gamma(27+10\gamma)\overline{E}_i + 2\gamma(4a(5+2\gamma) - (9+4\gamma)\overline{E}_j)}{8(3+\gamma)^2} \tag{A17}$$

$$\overline{E}_j = V_{jP}'(x) + \frac{\gamma(4a + \gamma\overline{E}_i + \gamma\overline{E}_j)}{2(3+\gamma)^2} + \frac{-2\gamma(27+10\gamma)\overline{E}_j + 2\gamma(4a(5+2\gamma) - (9+4\gamma)\overline{E}_i)}{8(3+\gamma)^2} \tag{A18}$$

By symmetry $V_{iP}(x) = V_{jP}(x) = V_P(x)$, we have

$$\overline{E}_i^* = \overline{E}_j^* = \frac{2\left(V_P'(x)(3+\gamma)^2 + a\gamma(7+2\gamma)\right)}{\gamma(18+5\gamma)} \tag{A19}$$

Substituting (A19) into HJB equation and we obtain

$$\rho V_P(x) = \frac{4(2a+V_P'(x))^2(3+\gamma)^2}{(18+5\gamma)^2} - \frac{2\left[6a\gamma V_P'(x)(3+\gamma)^2 + (V_P'(x))^2(3+\gamma)^2(9+4\gamma)\right]}{\gamma(18+5\gamma)^2} \\ - \frac{a^2\gamma(72+35\gamma+4\gamma^2)}{\gamma(18+5\gamma)^2} - dx(t) + V_P'(x)\left[\frac{4(V_P'(x)(3+\gamma)^2 + a\gamma(7+2\gamma))}{\gamma(18+5\gamma)} - \delta x(t)\right] \tag{A20}$$

We can also conjecture that the value function of region $i$ or region $j$ is linear in $x$, that is,

$$V_{iP}(x) = V_{jP}(x) = V_P(x) = A_P x + B_P \tag{A21}$$

Similar to the aforementioned analysis, we can determine $A_P$ and $B_P$ by substituting (A21) and $V_P'(x)$ into (A20), as follows,

$$A_P = -\frac{d}{\rho+\delta} \tag{A22}$$

$$B_P = \frac{2d^2(3+\gamma)^2(27+8\gamma) - 4ad\gamma(135+77\gamma+11\gamma^2)(\rho+\delta)}{\gamma(18+5\gamma)^2\rho(\rho+\delta)^2} + \frac{2a^2\gamma(144+83\gamma+12\gamma^2)}{\gamma(18+5\gamma)^2\rho} \tag{A23}$$

Then, substituting (A22) and (A23) into $V_P'(x)$, and referring to (A19), we can derive the equilibrium emission quotas, outputs and abatement levels:

$$q_{iP}^* = q_{jP}^* = \frac{2(3+\gamma)(2a(\rho+\delta)-d)}{(18+5\gamma)(\rho+\delta)} \tag{A24}$$

$$r_{iP}^* = r_{jP}^* = \frac{2(3d(3+\gamma) - a\gamma(\rho+\delta))}{\gamma(18+5\gamma)(\rho+\delta)} \tag{A25}$$

Substituting Equations (A24) and (A25) into (3), and by setting $\dot{x}_P(t) = 0$, the steady state of the pollution stock under permits trading policy can be obtained:

$$x_{PS}^* = \frac{4\left(a\gamma(7+2\gamma)(\rho+\delta) - d(3+\gamma)^2\right)}{\gamma\delta(18+5\gamma)(\rho+\delta)}$$

Then, solving the ordinary differential equation, we calculate the optimal trajectory of the pollution stock as in Proposition 3.

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
