# Peer review of "Optimal Environmental Policy in a Dynamic Transboundary Pollution Game: Emission Standards, Taxes, and Permit Trading"

_sustainability, doi:10.3390/su14159028_

Round 1

Reviewer 1 Report

This is an interesting topic, and has strong managerial implications. However, it has the following limitations.

 (1)It is noted that your manuscript needs careful editing by someone with expertise in technical English editing paying particular attention to English grammar, spelling, and sentence structure so that the goals and results of the study are clear to the reader.

(2) This topic is not new, many related papers have been published.

(3)Try to set the problem discussed in this paper in more clear, write one section to define the problem.

(4) With regard to the discussion of results, it would be advisable to link it to the evidence obtained in previous studies.

Author Response

Dear editor and reviewers:

Thank you for your comments concerning our manuscript entitled “Optimal environmental policy in a dynamic transboundary pollution game: emission standards, taxes, and permit trading” (sustainability-1814069). Those comments are all valuable and very helpful for revising and improving our paper, as well as the important guiding significance to our researches. We have studied comments carefully and have made correction which we hope meet with approval. The responses to reviewers’ comments are as following:

Key comments of the reviewer #1

Comment 1: It is noted that your manuscript needs careful editing by someone with expertise in technical English editing paying particular attention to English grammar, spelling, and sentence structure so that the goals and results of the study are clear to the reader.

Response: Considering the Reviewer’s valuable suggestion, we tried our best to improve the English presentation. We also used professional language editing service to improve and proofread the paper’s language. These changes in English expressions will not influence the content and framework of the paper.

Comment 2: This topic is not new; many related papers have been published.

Response: Thanks for your comments. Although the academic community has carried out a number of studies environ-mental policy and pollution control, there is still room for further expansion, as follows: (1) in the past, scholars have mainly studied the current situation and problems of environmental policy, but few scholars have studied the optimal environmental policy under transboundary pollution. This research expands and improves upon the existing research; (2) the existing studies did not explicitly consider inter-regional trade in goods or resources that cause pollution; (3) discussions on the evaluation of alternative environmental policies, such as emission taxes, emission standards, or marketable emission permits, are absent from the literature. These characteristics reflect the main variations between the present study and previous research.

Comment 3: Try to set the problem discussed in this paper in more clear, write one section to define the problem.

Response: Yes, you are right. First, we rewritten the introduction and added some key parts such as the purpose of our work, the significance of the work and our unique contribution towards the existing literature. Then, we also added the description of the paper structure at the end of the section. The revised introduction is as follows:

  1. Introduction

     According to the report of the global Environmental Performance Index (EPI) released by Yale University and Columbia University in 2018 (Wendling et al., 2018), the world is still far from achieving international environmental goals; the EPI score was reported to be only 46.16. This score reflects the current environmental condition and the implementation effects of environmental policies. The EPI reveals that the environmental quality of the world is improving as compared with the last few decades, but pollution remains severe, especially in developing countries and regions. Environ-mental pollution originates from the excessive discharge of pollutants by industrial firms. However, due to the pollutant attributes of externality and public goods, firms are reluctant to reduce their emissions or invest in emission abatement activities, resulting in market failure. Therefore, governments have adopted command-and-control or market-based environmental policies, such as emission standards, taxes, permit trading, etc., to govern the environment. As the most important means of pollution control, the above environmental policies have received extensive attention from academia and government [2-4]. Emission standards set quantitative limits on the permissible amount of specific air pollutants that may be released from specific sources over specific timeframes. As a command-and-control policy, if the pollutant emissions of enterprises exceed this standard, they will face serious environmental penalties [5]. Both emission taxes and emission permit trading are market-based environmental policies. The most significant difference between the two is the uncertainty of emission reduction and emission permit price. Emission tax is a price control policy, the tax rate is set by the policy makers, and the emission reduction is determined by the market. While the emission permit trading is a quantitative policy, the total emission is set by the policy maker, and the trading price is determined by the market mechanism. When the permit price or total emission is set at the place where the marginal abatement cost is equal to the marginal abatement income, the emission tax and emission permit trading can implement the same outcome under the completely competitive market conditions. Scholars have carried out long-term and systematic research on the comparison of the above policies. However, uncertainties in reality make all kinds of policies unable to achieve the theoretical optimal effect, and all kinds of policies need to be reasonably selected according to the specific environment [2],[6].

What’s more, a typical feature of pollution is that pollutants are released from one region and then migrate to another; this is called transboundary pollution, which often occurs in two or more neighboring regions. Many or all regions generate emissions, and many or all are also suffering from them. A celebrated example is the emissions of greenhouse gases (GHG), sulfur dioxide (SO2), and ozone, which cause global warming and environmental degradation [7]. Transboundary pollution not only destroys the ecological environment, but also endangers human health. Outdoor air pollution caused by PM2.5, SO2 and ozone lead to premature death, with more than 3million people worldwide every year [8]. Therefore, in view of the impact of neighboring regions, when the government regulates the environmental policies of local enterprises, it should not only consider the impact of local environmental pollution, but also consider the impact of the behavior of the government and enterprises in adjacent areas, which increases the difficulty of decision-making and may cause new changes in the optimal choice of environmental policies [9]. Most related studies conducted previously were limited to the optimal environmental policy of either local or domestic governments, and some scholars compared the emission taxes, standards, and permit trading policies under the conditions of different market situations and implementation environments [10-12]. However, these studies did not compare the environmental and social welfare effects of the three environmental policies, particularly the conditions of the optimal policy choice for the transboundary pollution problem, which is the main objective of the present study. The consideration of transboundary pollution is unique; for two or more adjacent regions, each region suffers not only from the damage caused by the pollution stock generated by local firms, but also from the pollution stock generated by firms in the neighboring region [13,14]. Therefore, the government of each region will also be affected by the neighboring region when implementing environmental policies.

Two adjacent regions are considered in this study. Each region has a representative firm that competes with the representative firm of the other region and sells homogeneous goods, and the firm in each region will emit pollutants during the production process. Consumers in the two regions can also freely purchase the products produced by the firms in the two regions. The goal of each government is to maximize the local social welfare under the constraint of the total pollution stock of the two regions. Moreover, the two regions are considered to suffer from the same environmental damage as a result of the total emissions. Thus, differential game models of the optimal choice problem are first formulated in the following three scenarios: emission standards, emission taxes, and emission permit trading. The feedback Nash equilibrium solutions of the total pollution stock and the social welfare of each region are then derived, and the three policies are compared. The results are illustrated with a numerical example.

The main contributions of this paper include the following: (1) Most of the existing studies are limited to the static situation, and focus on the relationship between a single government and enterprises. This paper studies the optimal choice of government environmental policy under the dynamic change of transboundary pollution and pollution capacity;this is the main expansion of and improvement to the existing research;(2)emission standards, emission taxes and emission permit trading were analyzed in our paper, it not only systematically compares the differences in environmental quality of emission standards, emission taxes and emission permits trading environmental policies, but also compares the impact on social welfare in each region, and analyzes the reasons for the differences. It has important practical significance for the control of transboundary pollution.

The main structure of this paper is as follows: the Section 1 of the paper introduces the research background of this paper and describes the purpose of the work and its significance. The Section 2 summarizes the relevant literatures and presents the contributions of scholars in this field and the re-search innovation of this paper. The Section 3 provides the game formulation between the government and firms under the condition of transboundary pollution. Subsequently, the Section 4 introduces the following scenarios: (i) the emission standards policy in Section 4.1; (ii) the emission tax policy in Section 4.2; (iii) the emission permit trading policy in Section 4.3. Section 4.4 compares two aspects of the three environ-mental policies, namely the total pollution stock and the social welfare benefits for the government in each region. In Section 5, the results are illustrated with a numerical example. Finally, the Section 6 concludes this study with a brief summary and suggests future research directions.

Comment 4: With regard to the discussion of results, it would be advisable to link it to the evidence obtained in previous studies.

Response: Yes, you are right. We linked our results to the evidence obtained in previous studies. For example, in Proposition 4, we added some previous studies and compared the results: Proposition 4 demonstrates that when the government in each region implements environmental policy under the condition of transboundary pollution, the total pollution stock of the two regions under the emission tax policy is the highest from the perspective of the environmental effect. Furthermore, among the three policies, the total pollution stock under the emission permit trading policy is the lowest. Our conclusion is partially consistent with the views of Ulph [26] and Yanase [44]. For example, Ulph considered the choice of environmental policy instruments (taxes or standards) in the context of a model of strategic international trade between countries. The results showed that if trade is modeled as a Stackelberg game, then both countries have higher producer surpluses under the emission standard policy. Yanase [44] considered the market structure of enterprises in two countries competing in a third country, and he analyzed the impacts of tax and standard policies adopted by the governments of the two countries on the total pollution stock. The results revealed that the emission standards policy is better than the emission tax policy in environmental quality. How-ever, these scholars did not participate in the analysis of emission permit trading policy, and did not compare the comprehensive impact on the environment and social welfare. In addition, they also did not analyze the deep-seated reasons for the above results. Our conclusion will fill these research gaps.

The reasons for this can be explained as follows. As presented in Sections 3.1-3.3, under the emission tax policy, when the government of the local region (i.e., region i) increases the emission tax rate (the policy tends to be strict), the firm in the local region will decrease its outputs, whereas the firm in the neighboring region (i.e., region j) will increase its outputs accordingly; this is the rent-shifting effect. Moreover, the increase of the emission tax rate by the local government will decrease the net emissions of the local region, but will lead to an increase of the net emissions in the neighboring region, resulting in policy leakage. This means that when the government attempts to improve the environmental quality by increasing the emission tax rate, “free-riding” behavior will occur in the neighboring region. For these reasons, when determining the emission tax rate, the government in each region will predict and consider the free-riding behavior of the other government. Therefore, the emission tax rates of both sides will be lower than the optimal level, which will lead to increases in the net emissions and pollution stock. Some literatures of environmental economics have been revealed that local governments tend to lower the standards of environmental regulation to attract scarce working capital and enterprises to enter the local area, resulting in inferior competition between governments and intensifying environmental pollution [51].

(You can see the detailed content in our paper of revised version. Thanks.)

Reviewer 2 Report

1. Abstract has inappropriate structure. I suggest to answer the following aspects: - general context - novelty of the work - methodology used (describe briefly the main methods or treatments applied) - main results and related interpretations. 2. Introduction: This section should briefly place the study in a wide context and emphasize why it is relevant carrying out the analysis. It should define the purpose of the work and its significance. In this perspective, this section is too succinct and fails to effectively point out the relevance of your contribution towards the existing literature. Moreover, the authors do not provide at the end of the section the description of the paper structure which is very useful for readers. 3. Literature Review: This chapter is important. The authors present a rather modest system of analysis that can be further improved. It would be useful to analyze more and new sources. Only 34 sources are analyzed in the work. In a journal of such a high scientific level as Sustainability, I think there are definitely not enough numbers. 4. The research methodology seems underdeveloped. Methods should be described in detail. I think the research procedure could be much more clearly described by means of a diagram also highlighting its potential and limit. The article is full of tables and figures, but I lack a more detailed explanation. 5. Results are not always linked to the methodology. Please define the relationship and relate your finding with the relevant literature. 5. There is no a discussion part  where authors should disclose their essential “discoveries”. I would suggest the authors to frame it as a "typical" conclusive section. Please provide limitation, future research needs as well as practical / policy implications. Also please check requirements as for example reference should cite like that [1].

Good luck

Author Response

Response to Reviewer 2 Comments

Dear editor and reviewers:

Thank you for your comments concerning our manuscript entitled “Optimal environmental policy in a dynamic transboundary pollution game: emission standards, taxes, and permit trading” (sustainability-1814069). Those comments are all valuable and very helpful for revising and improving our paper, as well as the important guiding significance to our researches. We have studied comments carefully and have made correction which we hope meet with approval. The responses to reviewers’ comments are as following:

Key comments of the reviewer #2

Comment 1: Abstract has inappropriate structure. I suggest to answer the following aspects: -

general context - novelty of the work - methodology used (describe briefly the main methods

or treatments applied) - main results and related interpretations.

Response: Yes, thanks very much. We rewritten the abstract by your kind advice as following:

Global environmental problems such as transboundary pollution and global warming have been recognized as major issues around the world. In practice, governments of all countries are actively exploring various environmental policies to control pollution. Especially in the context of transboundary pollution, the government needs to consider the impact of neighboring regions when formulating environmental policies. However, the above problems are less studied, to bridge this gap and especially aim at solving problems in existing practices, we consider a differential game model of transboundary pollution control to examine which policy is more effective in promoting the environmental quality and social welfare in a dynamic and accumulative global pollution context. Three alternative policy instruments, namely emission standards, emission taxes, and emission permit trading, are considered and compared. The results show that the social welfare of each region is the lowest and the total pollution stock is the highest under the emission tax policy due to the “rent-shifting,” “policy-leakage,” and “free-riding” effects. Moreover, the realized level of the environmental policy in the Nash equilibrium of the policy game is distorted away from the socially optimal level. The emission standards policy is found to be better than the emission tax policy and characterized by the rent-shifting effect but no policy-leakage effect. Moreover, the pollution stock of two regions is found to be the lowest and the social welfare is found to be the highest under the emission permit trading policy, which is not associated with any of the three effects. Finally, a numerical example is used to illustrate the results, and a sensitivity analysis is done in the steady state.

Comment 2: Introduction: This section should briefly place the study in a wide context and emphasize why it is relevant carrying out the analysis. It should define the purpose of the work and its significance. In this perspective, this section is too succinct and fails to effectively point out the relevance of your contribution towards the existing literature. Moreover, the authors do not provide at the end of the section the description of the paper structure which is very useful for readers.

Response: Thanks for your constructive suggestions. First, we rewritten the introduction and added some key parts such as the purpose of our work, the significance of the work and our unique contribution towards the existing literature. You can see the modified introduction in our paper of revised version. Thanks.

Then, we also added the description of the paper structure at the end of the section:

The main structure of this paper is as follows: the Section 1 of the paper introduces the research background of this paper and describes the purpose of the work and its significance. The Section 2 summarizes the relevant literatures and presents the contributions of scholars in this field and the research innovation of this paper. The Section 3 provides the game formulation between the government and firms under the condition of transboundary pollution. Subsequently, the Section 4 introduces the following scenarios: (i) the emission standards policy in Section 4.1; (ii) the emission tax policy in Section 4.2; (iii) the emission permit trading policy in Section 4.3. Section 4.4 compares two aspects of the three environmental policies, namely the total pollution stock and the social welfare benefits for the government in each region. In Section 5, the results are illustrated with a numerical example. Finally, the Section 6 concludes this study with a brief summary and suggests future research directions.

Comment 3: Literature Review: This chapter is important. The authors present a rather modest system of analysis that can be further improved. It would be useful to analyze more and new sources. Only 34 sources are analyzed in the work. In a journal of such a high scientific level as Sustainability, I think there are definitely not enough numbers.

Response: Thanks for your kind advice. Following your suggestion, we add more than 20 related papers and improved the Literature Review. You can See these literatures as in our paper.  

References

  1. Wendling, Z.A.; Emerson, J.W.; Esty, D.C.; Levy M.A.; de Sherbinin, A.; et al. Environmental Performance Index. Yale Center for Environmental Law & Policy, New Haven, CT, 2018.
  2. Weitzman, M.L. Prices vs. Quantities. Econ. Stud. 1974, 41, 477–491.
  3. Newell, R.G.; Pizer, W.A. Regulating Stock Externalities under Uncertainty. Environ. Econ. Manage. 2003, 45, 416–432.
  4. Weber, T.A.; Neuhoff, K. Carbon Markets and Technological Innovation. Environ. Econ. Manage. 2010, 60, 115–132.
  5. Arguedas, C.; Cabo, F.; Martín-Herrán, G. Optimal Pollution Standards and Non-compliance in a Dynamic Framework. Resour. Econ. 2017, 68, 537–567.
  6. Martín-Herrán, G.; Rubio, S. J. Optimal Environmental Policy for a Polluting Monopoly with Abatement Costs: Taxes versus Standards. Model. Assess. 2018, 23, 671–689.
  7. Jørgensen, S.; Martín-Herrán, G.; Zaccour, G. Dynamic Games in the Economics and Management of Pollution. Model. Assess. 2010, 15, 433–467.
  8. Zhang, Q.; Jiang, X.; Tong, D.; Davis, S. J.; Zhao, H.; Geng, G.; ... Guan, D. Transboundary Health Impacts of Transported Global Air Pollution and International trade, Nature 2017, 7647, 705–709.
  9. Ambec, S.; Coria, J. Prices vs Quantities with Multiple Pollutants. Environ. Econ. Manage. 2013, 66, 123–140.
  10. Requate, T. Pollution Control in a Cournot Duopoly via Taxes or Permits. Econ. 1993, 58, 255–291.
  11. Kato, K. Emission Quota versus Emission Tax in a Mixed Duopoly. Econ. Policy Stud. 2011, 13, 43–63.
  12. Wirl, F. Taxes versus Permits as Incentive for the Intertemporal Supply of a Clean Technology by a Monopoly. Energy Econ. 2014, 36, 248–269.
  13. Dockner, E. J.; Van Long, N. International Pollution Control: Cooperative versus Noncooperative Strategies. Environ. Econ. Manage. 1993, 25, 13–29.
  14. Van Long, N. A Survey of Dynamic Games in Economics, World Scientific, Singapore, 2010; PP. 71-104.
  15. Benchekroun, H.; Van Long, N. Efficiency Inducing Taxation for Polluting Oligopolists. Public Econ. 1998, 70, 325–342.
  16. Storrøsten, H.B. Prices versus Quantities: Technology Choice, Uncertainty and Welfare. Resour. Econ. 2014, 59, 275–293.
  17. Masoudi, N.; Zaccour, G. Emissions Control Policies under Uncertainty and Rational Learning in a Linear-state Dynamic Model. Automatica 2014, 50, 719–726.
  18. Masoudi, N. Environmental Policies in the Presence of More Than One Externality and of Strategic Firms. Environ. Plan. Manag. 2022, 65, 168–185.
  19. Garcia, A.; Leal, M.; Lee, S.H. Optimal Policy Mix in an Endogenous Timing with a Consumer-Friendly Public Firm. Bull. 2018, 38, 1438–1445.
  20. Feenstra, T.; Kort, P. M.; de Zeeuw, A. Environmental Policy Instruments in an International Duopoly with Feedback In-vestment Strategies. Econ. Dyn. Control. 2001, 25, 1665–1687.
  21. Hoel, M.; Karp, L. Taxes versus Quotas for a Stock Pollutant. Resour. Energy Econ. 2002, 24, 367–384.
  22. Lee, S. H.; Park, S. H. Tradable Emission Permits Regulations: The Role of Product Differentiation. J. Bus. Econ. 2005, 4, 249–261.
  23. Garcia, A.; Leal, M.; Lee, S. H. Time-inconsistent Environmental Policies with a Consumer-friendly Firm: Tradable Permits versus Emission Tax. Rev. Econ. Financ. 2018, 58, 523–537.
  24. Feichtinger, G.; Lambertini, L.; Leitmann, G.; et al. R&D for Green Technologies in a Dynamic Oligopoly: Schumpeter, Arrow and inverted-U’s. J. Oper. Res. 2016, 249, 1131–1138.
  25. Moner-Colonques, R.; Rubio, S. The Timing of Environmental Policy in a Duopolistic Market. Agrar. Recur. Nat. 2015, 15, 11–40.
  26. Ulph, A. The Choice of Environmental Policy Instruments and Strategic International Trade; In Conflicts and Cooperation in Managing Environmental Resources, Springer, Berlin, Heidelberg.1992; PP. 111–132.
  27. Lai, Y.; Hu, C. Trade Agreements, Domestic Environmental Regulation, and Transboundary Pollution. Energy Econ. 2008, 30, 209–228.
  28. Glachant, M.; Ing, J.; Nicolai, J. P. The Incentives for North-south Transfer of Climate-mitigation Technologies with Trade in Polluting Goods. Resour. Econ. 2017, 66, 535–456.
  29. Dockner, E. J.; Jorgensen, S.; Van Long, N.; Sorger, G. Differential Games in Economics and Management Science; Cam-bridge University Press: Cambridge, UK, 2000; PP, 37–80.
  30. Dockner, E. J.; Van Long, N. International Pollution Control: Cooperative versus Noncooperative Strategies. Environ. Econ. Manage. 1993, 25, 13–29.
  31. Jørgensen, S.; Zaccour, G. Time Consistent Side Payments in a Dynamic Game of Downstream Pollution. Econ. Dyn. Control. 2001, 25, 1973–187.
  32. Jørgensen, S.; Martín-Herrán, G.; Zaccour, G. Agreeability and Time Consistency in Linear-state Differential games. Optim. Theory Appl. 2003, 119, 49–63.
  33. Breton, M.; Sbragia, L.; Zaccour, G. A Dynamic Model for International Environmental Agreements. Resour. Econ. 2010, 45, 25–48.
  34. Li, S. A Differential Game of Transboundary Industrial Pollution with Emission Permits Trading. Optim. Theory Appl. 2014, 163, 642–659.
  35. Bertinelli, L.; Camacho, C.; Zou, B. Carbon Capture and Storage and Transboundary Pollution: A Differential Game Approach. J. Oper. Res. 2014, 237, 721–728.
  36. El Ouardighi, F.; Kogan, K.; Gnecco, G.; Sanguineti, M. Transboundary Pollution Control and Environmental Absorption Efficiency Management. Oper. Res. 2020, 287, 653–681.
  37. Yeung, D. W.; Petrosyan, L.A. A Cooperative Stochastic Differential Game of Transboundary Industrial Pollution. Automatica, 2008, 44, 1532–1544.
  38. Xu, H.; Tan, D. Optimal Abatement Technology Licensing in a Dynamic Transboundary Pollution Game: Fixed Fee versus Royalty. Econ. 2019. in press.
  39. Marsiglio, S.; Masoudi, N. Transboundary Pollution Control and Competitiveness Concerns in a Two-country Differential Game. Model. Assess, 2022, 27, 105–118.
  40. Huiquan, Li.; Genlong, Guo. Dynamic Decision of Transboundary Basin Pollution under Emission Permits and Pollution Abatement. Physica A 2019, 532, 121869–121869.
  41. de Frutos, J.; Gatón, V.; López-Pérez, P.M.; et al. Investment in Cleaner Technologies in a Transboundary Pollution Dynamic Game: A Numerical Investigation. Games Appl. 2022, in press.
  42. de Frutos, J.; Martín-Herrán, G. Spatial vs. Non-spatial Transboundary Pollution Control in a Class of Cooperative and Non-cooperative Dynamic Games[J]. J. Oper. Res. 2019, 276, 379–394.
  43. Yanase, A. Dynamic Games of Environmental Policy in a Global Economy: Taxes versus Quotas. Int. Econ. 2007, 15, 592–611.
  44. Yanase, A. Global Environment and Dynamic Games of Environmental Policy in an International Duopoly. Econ. 2009, 97, 121–140.
  45. Yanase, A.; Kamei, K. Dynamic Game of International Pollution Control with General Oligopolistic Equilibrium: Neary Meets Dockner and Long. Games Appl. 2022, in press.
  46. Menezes, F. M.; Pereira, J. Emissions Abatement R&D: Dynamic Competition in Supply Schedules. Public. Econ. Theory. 2017, 19, 841–859.
  47. Benchekroun, H.; Chaudhuri, A.R. Transboundary Pollution and Clean Technologies. Energy Econ. 2014, 36, 601–619.
  48. Martín-Herrán, G.; Rubio, S.J. On Coincidence of Feedback and Global Stackelberg Equilibria in a Class of Differential Games. J. Oper. Res, 2021, 293, 761–772.
  49. Cheng, K.F.; Tsai, C.S.; Hsu, C.C.; et al. Emission Tax and Compensation Subsidy with Cross-industry Pollution. Sustainability 2019, 11, 998.
  50. Conrad, K. Taxes and Subsidies for Pollution-intensive Industries as Trade Policy. Environ. Econ. Manage. 1993, 25, 121–135.
  51. Konisky, D.M. Regulatory Competition and Environmental Enforcement: Is There a Race to the Bottom? J. Polit. Sci. 2007, 51, 853–872.
  52. Lambertini, L.; Poyago-Theotoky, J.; Tampieri, A. Cournot Competition and “Green” Innovation: An Inverted-U Relation-ship. Energy Econ. 2017, 68, 116–123.
  53. Helm, D. The European Framework for Energy and Climate Policies. Energy Policy 2014, 64, 29–35.
  54. Burtraw, D.; McCormack, K. Consignment Auctions of Free Emissions Allowances. Energy Policy 2017, 107, 337–344.
  55. Khezr, P.; MacKenzie, I.A. Consignment Auctions[J]. Environ. Econ. Manage. 2018, 87, 42–51.

Comment 4: The research methodology seems underdeveloped. Methods should be described in detail. I think the research procedure could be much more clearly described by means of a diagram also highlighting its potential and limit. The article is full of tables and figures, but I lack a more detailed explanation.

Response: Thanks for the editor’s kind advice. We modified the Section 2 into two parts: 3.1---- Parameter Description and Assumptions and 3.2----Methods. You can see the methodology of this paper as follows:

3.2. Methods

3.2.1 Differential game

Differential games have distinct advantages in order to represent the interdependencies among time, strategic behavior and participants in mathematical models in the fields of environmental economics and optimal pollution control. Firstly, an ad-vantage of differential games in applications to problems of pollution is the opportunity to model damage caused by stock of accumulated pollution (Jørgensen et al. 2010). So, one of the key assumptions of this paper is emissions by either region con-tribute to the stock of pollution and the two regions face the same pollution stock. Secondly, the strategy structure of the differential game solution reflects the interaction of the behaviors between the participants. By establishing the Hamilton-Jacobi-Bellman equation (HJB), the Markov perfect Nash equilibrium (MPNE) is obtained. Under this equilibrium, the participants not only consider the dynamic changes of state variables, but also adjust their own strategies under the decision-making choices of other participants, and finally achieve time consistency and sub-game perfection. However, for multiple state equations or nonlinear situations, the differential game cannot get its analytical solution, so numerical analysis can be used to verify the results, which is its limitation.

3.2.2 Stackelberg Game

    The Stackelberg leadership model is a strategic game in economics in which the leader firm moves first and then the follower firms move sequentially. In our paper, it is assumed that the government has more information about the total environmental quality and its impact on local welfare than do the firms. Specifically, the pollution stock and environmental damage of each region in Eq. (3) can be observed by the governments (Leaders), but it cannot be judged by the firms (Followers). Therefore, the firm cannot predict policy information, such as the emission standards, emission tax rates, or initial emission permits set by the government. These assumptions indicate that the local government can obtain information about the firm’s feedback strategy and the damage caused by the total pollution in the two regions to formulate the optimal temporal path under each environmental policy to maximize social welfare. In particular, a three-stage Stackelberg game is considered in which (1) in stage one, the government in each region sets the standards/taxes/quotas to maximize social welfare, (2) in stage two, firms determine the abatement level, and (3) in stage three, market competition occurs (firms determine output). We can use backward induction to solve this problem and guarantee the equilibrium solution concept is perfect subgame equilibrium [48,49].

Comment 5: Results are not always linked to the methodology. Please define the relationship and relate your finding with the relevant literature.

Response: Thanks for the reviewer’s kind advice. We defined the relationship between results and methodology, and also related our finding with the previous literatures.

 First, this paper used differential games to analyze the optimal environmental policy under transboundary pollution, an advantage of differential games in applications to problems of pollution is the opportunity to model damage caused by stock of accumulated pollution (Jørgensen et al. 2010). So, one of the key assumptions of this paper is emissions by either region contribute to the stock of pollution and the two regions face the same pollution stock. We used dynamic equation of total pollution stock to describe this methodology. In our results such as Proposition 1, Proposition 2 and Proposition 3, we described the total pollution stock trajectory of both regions and used numerical analysis to express our methodology. Furthermore, this paper used Stackelberg and we explained this methodology in Section 3.2.2.

 We also added the Section 3.2----Methods to better describe the relationship between methodology and our model.

In addition, we also related our finding with the previous literatures. For example, in Proposition 4, we added some previous studies and compared the results: Proposition 4 demonstrates that when the government in each region implements environmental policy under the condition of transboundary pollution, the total pollution stock of the two regions under the emission tax policy is the highest from the perspective of the environmental effect. Furthermore, among the three policies, the total pollution stock under the emission permit trading policy is the lowest. Our conclusion is partially consistent with the views of Ulph [26] and Yanase [44]. For example, Ulph considered the choice of environmental policy instruments (taxes or standards) in the context of a model of strategic international trade between countries. The results showed that if trade is modeled as a Stackelberg game, then both countries have higher producer surpluses under the emission standard policy. Yanase [44] considered the market structure of enterprises in two countries competing in a third country, and he analyzed the impacts of tax and standard policies adopted by the governments of the two countries on the total pollution stock. The results revealed that the emission standards policy is better than the emission tax policy in environmental quality. How-ever, these scholars did not participate in the analysis of emission permit trading policy, and did not compare the comprehensive impact on the environment and social welfare. In addition, they also did not analyze the deep-seated reasons for the above results. Our conclusion will fill these research gaps.

The reasons for this can be explained as follows. As presented in Sections 3.1-3.3, under the emission tax policy, when the government of the local region (i.e., region i) increases the emission tax rate (the policy tends to be strict), the firm in the local region will decrease its outputs, whereas the firm in the neighboring region (i.e., region j) will increase its outputs accordingly; this is the rent-shifting effect. Moreover, the increase of the emission tax rate by the local government will decrease the net emissions of the local region, but will lead to an increase of the net emissions in the neighboring region, resulting in policy leakage. This means that when the government attempts to improve the environmental quality by increasing the emission tax rate, “free-riding” behavior will occur in the neighboring region. For these reasons, when determining the emission tax rate, the government in each region will predict and consider the free-riding behavior of the other government. Therefore, the emission tax rates of both sides will be lower than the optimal level, which will lead to increases in the net emissions and pollution stock. Some literatures of environmental economics have been revealed that local governments tend to lower the standards of environmental regulation to attract scarce working capital and enterprises to enter the local area, resulting in inferior competition between governments and intensifying environmental pollution [51].

(You can see the detailed content in our paper of revised version. Thanks.)

Comment 6: There is no a discussion part where authors should disclose their essential “discoveries”. I would suggest the authors to frame it as a "typical" conclusive section. Please provide limitation, future research needs as well as practical / policy implications. Also please check requirements as for example reference should cite like that [1].

Response: Yes, thanks for your advice. First, we revised the Section 6 (Conclusion) and divided into two parts. The first part discloses our essential “discoveries” and provides some policy recommendations especially for government. The second part gives the limitations and prospects. The modified section is as follows:

  1. Conclusion

6.1. Conclusions

The environmental policy literature usually concentrates on a single region or country. However, a substantial amount of environmental pollution in the world is caused not only by firms in a domestic region, but also by firms in neighboring regions. Therefore, governments will also be affected by neighboring regions when making environmental policies. In this paper, a transboundary pollution problem was investigated in which there is a Stackelberg game between firms and their local governments, and the government in each region chooses three different environmental policies, namely emission standards, emission taxes, and emission permit trading. The feedback Nash equilibrium solutions in the three scenarios were derived, and the three environmental policies were compared from the perspectives of environmental quality and social welfare. The results revealed that: (1) when the emission abatement cost coefficient of the firms in the two regions is the same, the two regions suffer the same environmental damage. Due to the rent-shifting, policy-leakage, and free-riding effects, the total pollution stock was found to be the highest and the social welfare of each region was found to be the lowest under the emission tax policy. The emission standards policy was found to be better than the emission tax policy due to the presence of the rent-shifting effect but the absence of the policy-leakage effect. Finally, the pollution stock of the two regions was found to be the lowest and the social welfare was found to be the highest under the emission permit trading policy, which does not have any of the three effects; (2) the dynamic trajectories of the equilibrium results and the sensitivity analysis of the parameters in the steady state are discussed numerically, the basic results reveal that when the initial pollution stock of the total regions is higher (lower) than the pollution stock in the steady state, the trajectory first decreases (rises) and converges to the steady level with the passage of time. In addition, the trajectory of the social welfare curve is similar to the pollution stock. With the initial pollution stock higher or lower than that the steady state, the social welfare of each region will increase or decrease to the steady-state level accordingly; (3) We analyzed the impact of changes in parameters such as environmental damage coefficient, market size, and natural decay rate on the total pollution stock and social welfare in each region under steady-state conditions.

In order to fully recognize the significance of the environmental policy under transboundary pollution and improve the total environmental quality, the government should mainly focus on the following four aspects: first, for some countries or regions with serious cross-border pollution, the implementation of regional emissions trading policies may be a better environmental governance model. For example, in order to alleviate greenhouse gas emissions, the EU implemented European Union Emissions Trading System (EUETS). EUETS in 2005, which has a significant inhibitory effect on the emission of carbon dioxide and other air pollutants in the European Union [48]. Second, although the research conclusion of this paper believes that the emission tax has the worst effect on environment and welfare, for some developing countries such as China or India, the level of marketization is relatively low, so it can still be combined with the emission tax policy and the emission standard policy to control the pollution.

6.2. Limitations and Prospects

The present work discussed the differential game problem of transboundary pol-lution. However, the difference in the market structure was not considered, which may also yield interesting and comparable results. For example, it would be interesting to know how the environmental policy would change if the market structure is difference. In addition, the results are based on the traditional set-up, such as the costate equation for pollution stock and the pollutants damage. Especially in recent years, a single source of emissions is typically comprised of multiple pollutants that also lead to regional and global negative externalities in reality. Finally, this paper assumes that the government utilizes a grandfathering approach for initial allocating of costless permits to the firms. But more and more governments are now using auctions to deal with the initial allocation of emission rights [53, 54]. These issues represent possible extensions of the present study.

Finally, for problem of “check requirements as for example reference should cite like that [1].”, we used the Sustainability Microsoft Word template file and modified all the citation format of the references.

Round 2

Reviewer 2 Report

Accept in present form